# Gated Graph Attention Networks with Multichannel Fusion for Disease Comorbidity Prediction

## Abstract

The co-occurrence of multiple diseases, or comorbidity, significantly complicates clinical management and worsens patient outcomes. Comorbidity is believed to arise from genetic mutations functionally connected through protein-protein interactions (PPIs) within the human interactome. Unraveling these intricate PPI networks is essential for understanding disease progression and addressing the challenges posed by comorbid conditions. In this study, we propose a novel Gated Graph Attention Network (GGAT) framework tailored for disease comorbidity prediction, which differs from the state-of-the-art Graph Transformer with Subgraph Positional Encoding in three key aspects: (1) it applies attention over local neighbors rather than global pairwise attention among all protein nodes, enabling more biologically meaningful aggregation; (2) it incorporates a gating mechanism to adaptively regulate information flow and enhance representation learning for comorbidity prediction; and (3) it introduces a multichannel fusion strategy that integrates connectivity based and disease association based embeddings, both of which have been shown to be important for disease comorbidity prediction. Experimental results on the benchmark dataset demonstrate that GGAT significantly outperforms the Transformer baselines across all metrics (AUROC, AUPRC, accuracy, and MCC), with the multichannel gated fusion variant achieving the best overall performance. These findings highlight the importance of integrating complementary biological features through graph structure and indicate that the proposed GGAT provides a generalizable graph learning framework applicable beyond disease comorbidity prediction.

## 1 Introduction

Comorbidity, the simultaneous presence of multiple medical conditions in an individual (Goodell et al., 2011), is a major factor shaping disease management, treatment strategies, and patient outcomes. Analyzing comorbidity patterns reveals complex disease relationships and shared molecular pathways (Guo et al., 2018; Keezer et al., 2016; Sarfati et al., 2016; Renzi et al., 2019; Sanyaolu et al., 2020). Genes influence disease phenotypes through the proteins they encode, and these proteins interact extensively with one another both within and outside the cell, forming protein–protein interaction (PPI) networks that provide a natural framework for studying comorbidity. The Human Interactome (HI) (Menche et al., 2015) is a systematic map of human PPIs, compiled from experimentally validated interactions and curated predictions. Disease-associated subgraphs within the HI (disease modules) provide valuable insights that enable systematic studies of disease comorbidity (Menche et al., 2015; Lauder et al., 2023). Machine learning methods commonly map protein–protein interaction (PPI) networks, including the Human Interactome, into low-dimensional node representations that capture structural and biological properties (Lian et al., 2019; Yang et al., 2020b; Madeddu et al., 2022; Ghosh et al., 2023), which in turn provide the basis for modeling disease comorbidity (Akram & Liao, 2019; Qin & Liao, 2025a;b).

Among recent comorbidity prediction studies, the Biologically Supervised Embedding (BSE) (Qin & Liao, 2025a) study revealed that both protein node connectivity within the interactome and disease–protein association patterns provide biologically meaningful signals for disease comorbidity prediction. Building on this insight, the state-of-the-art (SOTA) baseline, Graph Transformer

with Disease Subgraph Positional Encoding (TSPE) (Qin & Liao, 2025b), incorporates these two sources of information into a Transformer framework. Specifically, Node2Vec (Grover & Leskovec, 2016) embeddings are used to encode connectivity, while disease association information is injected through subgraph positional encoding. TSPE constructs SPE by adding Laplacian positional encoding (Dwivedi & Bresson, 2020) to the connectivity embedding and then concatenating a low-dimensional projection of Graph Encoder Embedding (GEE) (Shen et al., 2022; Qin & Shen, 2024), enabling the model to capture both clustering structure and disease-subgraph membership. The final SPE-enhanced embedding is fed into a Transformer encoder–decoder, where self-attention models relations within each disease module and cross-attention captures interactions across the two disease modules. For any disease pair (A, B), the encoder processes all protein nodes linked to disease A, and the decoder processes those linked to disease B, enabling multi-head attention that performs self-attention within individual disease modules and cross-attention between the two modules. By jointly leveraging connectivity and association signals, TSPE achieves strong predictive performance in disease comorbidity prediction.

Although TSPE substantially improves predictive performance over earlier methods, three key limitations remain. First, the Transformer's global attention mechanism computes pairwise interactions among all protein nodes within and between the two disease modules, regardless of whether a true PPI edge exists, limiting its ability to exploit the intrinsic sparsity of the interactome. Second, TSPE's subgraph positional encoding is inherently constrained in how much disease information it can incorporate: because disease label information is injected by concatenating a low-dimensional GEE projection to the input embedding, the amount of disease signal that TSPE can represent is tightly limited. Increasing this dimensionality would enlarge the Transformer input and thus significantly raise the computational cost of each self- and cross-attention block, creating an unavoidable dimensionality trade-off. Third, TSPE's design lacks flexibility for integrating additional biological features. Any new signals identified in the future would be difficult to incorporate, since the design is limited to combining only connectivity and disease information.

Graph Attention Networks (GATs) (Vrahatis et al., 2024; Velickovic et al., 2017) address the Transformer's limited use of graph topology by learning attention weights over true PPI neighbors and aggregating informative local signals, which is advantageous in noisy interactomes. To improve the stability and regulation of feature propagation, we incorporate gating mechanisms, which have been shown in both sequence models and graph neural networks (Dey & Salem, 2017; Dwivedi & Bresson, 2020; Sit et al., 2021) to effectively control how information is transmitted across layers. We also introduce a multichannel fusion strategy that integrates two biological signals, connectivity protein embeddings and disease–protein association embeddings, while overcoming the constraints of TSPE. Instead of concatenating disease label information into a single high-dimensional input embedding, each signal is processed in its own channel and later fused through a lightweight gated module. This design also avoids TSPE's limitations and provides greater flexibility within the node-aligned setting by allowing additional learned representations that map to the same protein nodes, while not constituting a general multimodal integration framework. Building on these insights and addressing the limitations of TSPE, we propose a Gated Graph Attention Network (GGAT) framework for disease comorbidity prediction. Our key contributions are:

- **Local biological aggregation.** We apply graph attention to local PPI neighborhoods rather than global pairwise attention as in TSPE, enabling more biologically meaningful aggregation within the interactome.
- **Gated information flow.** We incorporate a gating mechanism into GAT layers to adaptively regulate information flow, thereby enhancing representation learning for comorbidity prediction.
- **Flexible multichannel integration.** We introduce a multichannel fusion strategy that treats each input signal as an independent channel, enabling the architecture to incorporate additional useful learned representations when such data are available. This architecture naturally supports the integration of additional learned node representations in future settings, while the present study focuses strictly on these two sources consistent with the interactome benchmark.

## 2 RELATED WORK

### 2.1 GRAPH NEURAL NETWORKS AND ATTENTION-BASED MESSAGE PASSING

Recent years have seen a shift from traditional graph methods to graph neural networks (GNNs) (Zitnik et al., 2018; Fout et al., 2017; Jha et al., 2022; Yang et al., 2020a; Valous et al., 2024) for mod-

eling complex biological networks. GNNs leverage graph topology for representation learning. In fact, the Transformer (Vaswani et al., 2017) can be interpreted as a special case of GNNs operating on a fully connected graph, where edges are implicitly constructed between all input nodes. Unlike GNNs that propagate information along the true graph edges, the Transformer defines its own graph through the attention mechanism, which may overlook meaningful, task-specific neighborhood patterns. Among GNNs, Graph Convolutional Networks (GCNs) (Zhang et al., 2019; Kipf, 2016) and Graph Attention Networks (GATs) (Vrahatis et al., 2024; Velickovic et al., 2017) are well-suited to address this limitation. While GCNs aggregate information uniformly from neighbors, GATs extend this approach by learning attention weights, enabling the model to emphasize the most informative connections. This selective attention mechanism enhances GATs' ability to capture complex relationships within the graph, making it particularly useful in scenarios where the significance of node connections varies. Additionally, GATs are more flexible than GCNs, as they learn attention weights instead of relying on fixed Laplacian-based averaging. This allows GATs to adaptively focus on informative neighbors, which can be beneficial in noisy biological networks such as HI. These properties make GAT a natural foundation for our GGAT framework, which enhances attention-based message passing with a gating mechanism.

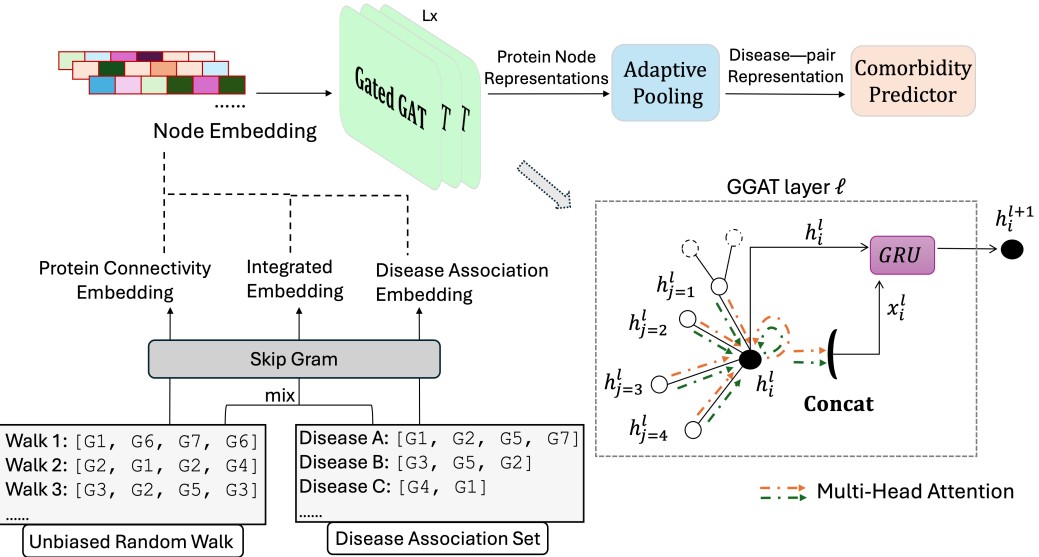

Figure 1: GGAT single-channel framework with embedding variants and a GGAT-layer inset. Different input embeddings define different channels: protein connectivity embeddings yield GGAT-Connect, disease-association embeddings yield GGAT-Disease, and integrated embeddings correspond to GGAT-EmbedFusion. Sequences of protein nodes from random walks on the interactome, such as [G1, G6, G7, G6], and sets of disease–associated proteins, such as [G4, G1], are both treated as sentences in a Skip-gram model to produce the initial node embeddings. The inset illustrates one GGAT layer combining multi-head attention with a GRU gate.

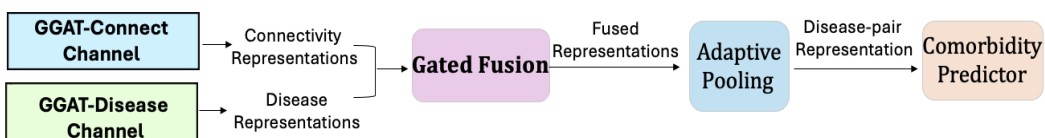

Figure 2: Overview of the GatedFusion model, where connectivity-based and disease-based node representations from two GGAT channels are integrated through a gated fusion layer, followed by adaptive pooling and a two-layer MLP predictor (same architecture as in the single-channel models) to estimate disease comorbidity.

## 2.2 GATING MECHANISMS IN GRAPH REPRESENTATION LEARNING

Gating mechanisms have long been established in recurrent neural networks (RNNs) such as Long Short-Term Memory (LSTM) (Sherstinsky, 2020) and Gated Recurrent Unit (GRU) (Dey & Salem, 2017).By regulating information flow, these models achieve significantly better trainability, stability, and performance than vanilla RNNs (Elman, 1990), while mitigating gradient-related issues (Noh, 2021). Inspired by these successes, gating mechanisms have also been incorporated into graph neural networks (GNNs), leading to notable improvements in their ability to capture complex dependencies (Ghasemzadeh et al., 2023; Wang et al., 2022; Lu et al., 2020). Dwivedi & Bresson (2020) systematically compared Generalized Transformer Networks (GT) with GCN (Kipf, 2016), GAT (Velickovic et al., 2017), and Gated GCN (Bresson & Laurent, 2017) across multiple benchmarks (Dwivedi et al., 2023). GT leverages fully connected attention and positional encoding to capture domain knowledge, outperforming GCN and GAT but falling short of Gated GCN. The superior performance of Gated GCN highlights the value of integrating true graph structure with gated feature aggregation to better regulate information flow.

## 2.3 DISEASE COMORBIDITY ANALYSIS AND BROADER DISEASE MODELING

Interactome-based disease–disease analysis is rooted in the broader network medicine framework (Pawson & Linding, 2008; Zanzoni et al., 2009; Barabási et al., 2011), and the seminal work by Menche et al. (Menche et al., 2015) demonstrated how incomplete interactome maps can be used to uncover disease modules and quantify disease–disease relationships including comorbidity. Since then, the Menche human interactome has been widely used in downstream network-medicine studies, including tissue-specific disease module mapping (Kitsak et al., 2016), interactome-based approaches to human disease (Caldera et al., 2017), and mechanism-level disease–disease similarity analysis (Sharma et al., 2015). Our work follows this line of research, focusing specifically on interactome-based disease comorbidity prediction, for which all prior models evaluated on the same benchmark include embedding-based method (Akram & Liao, 2019; Qin & Liao, 2025a) and graph transformer frameworks (Qin & Liao, 2025b). Beyond interactome-based methods, broader comorbidity analyses have incorporated a variety of heterogeneous biomedical data modalities, including electronic health records (EHR) (Kirk et al., 2019), genotype–phenotype information (Gasparini, 2018), GWAS-based associations (Cano-Gamez & Trynka, 2020), biological pathway–based information (Lee et al., 2008; Lei et al., 2023; Sánchez-Valle et al., 2020), knowledge-graph–based inference (Biswas et al., 2019), and multi-omics data (Barh et al., 2021). Other computational methods widely used for general disease analysis, although not for disease–disease comorbidity analysis, include tensor-factorization models (Wang et al., 2021; Zhao et al., 2019), gene co-expression network analysis (Van Dam et al., 2018), and single-cell transcriptomic profiling (Tian et al., 2022). We list these approaches here only to position interactome-based comorbidity modeling within the broader landscape of computational disease analysis.

## 3 METHOD

The proposed GGAT framework is designed with two levels of representation learning. Section 3.1 introduces the GGAT single-channel model, which applies a gating mechanism between graph-attention layers to regulate information flow. Section 3.2 then extends this design with two alternative fusion strategies: (i) EmbedFusion, which integrates connectivity and disease-association information at the embedding stage within a single channel, and (ii) GatedFusion, which adopts a dual-channel architecture and adaptively combines the outputs of two GGAT channels through a gated fusion mechanism. The following subsections describe each component in detail. For clarity, bias terms are omitted in all subsequent equations.

## 3.1 GGAT SINGLE-CHANNEL

As shown in Fig. 1, the GGAT single-channel extends the standard Graph Attention Network (GAT) by inserting a gating mechanism after each GAT layer to regulate information flow. Input node embeddings are propagated through stacked GGAT layers to obtain protein-level representations, which are then aggregated via adaptive pooling into disease-pair embeddings and passed to a predictor for binary comorbidity classification.

At each GGAT layer $l$, we adopt the standard multi-head GAT operation with $K$ parallel attention heads (we use $K = 8$ in our implementation) (Velickovic et al., 2017). For node $i$, each head computes attention coefficients over its neighbors and aggregates them into head-specific representations, which are then concatenated. We follow the original GAT formulation for these updates and include the explicit equations in Appendix B.1 for completeness.

Since stacked GAT layers tend to accumulate noise and exhibit unstable attention behavior on noisy biological networks, we augment each GAT layer with a GRU-style gating module (Dey & Salem, 2017). The GAT output $\boldsymbol{x}_i^{(l)}$ represents newly aggregated neighborhood information, while $\boldsymbol{h}_i^{(l)}$ denotes the node's representation produced by the previous layer. The gating mechanism integrates these two sources of information and regulates how much attention-derived signal is incorporated at each depth, thereby improving stability and mitigating noise and over-smoothing on the interactome. The gates are computed from the concatenated vector $[\boldsymbol{h}_i^{(l)} \,\|\, \boldsymbol{x}_i^{(l)}]$, which is algebraically equivalent to the canonical split form $\boldsymbol{U}^{(l)}\boldsymbol{h}_i^{(l)} + \boldsymbol{V}^{(l)}\boldsymbol{x}_i^{(l)}$(see Appendix A). Importantly, unlike the widely used temporal GRU formulations (e.g., Cho et al. 2014; Dey & Salem 2017; Zhao et al. 2018), which interpolate between $h_{t-1}$ and $x_t$ using separate $W$ and $U$ projections, our gating module is not a recurrent unit. Instead, our GRU-style gate provides a layer-wise feature fusion mechanism tailored for graph neural networks. The gates jointly condition on the concatenated vector $[\boldsymbol{h}_i^{(l)} \,\|\, \boldsymbol{x}_i^{(l)}]$, allowing the model to adaptively combine inherited node representations with newly produced attention messages—a structural role that differs from classical GRUs. The reset, update, candidate, and next-state computations follow the GRU functional form:

$$\boldsymbol{r}_i^l = \sigma\big(\boldsymbol{W}_r^l[\boldsymbol{h}_i^l \,\|\, \boldsymbol{x}_i^l]\big), \tag{1}$$

$$\boldsymbol{z}_i^l = \sigma\big(\boldsymbol{W}_z^l[\boldsymbol{h}_i^l \,\|\, \boldsymbol{x}_i^l]\big), \tag{2}$$

$$\tilde{\boldsymbol{h}}_i^l = \tanh\big(\boldsymbol{W}_h^l[\boldsymbol{h}_i^l \,\|\, (\boldsymbol{r}_i^l \odot \boldsymbol{x}_i^l)]\big), \tag{3}$$

$$\boldsymbol{h}_i^{l+1} = (1 - \boldsymbol{z}_i^l) \odot \boldsymbol{h}_i^l + \boldsymbol{z}_i^l \odot \tilde{\boldsymbol{h}}_i^l. \tag{4}$$

Here, $\boldsymbol{W}_r^l, \boldsymbol{W}_z^l, \boldsymbol{W}_h^l$ are trainable matrices, $\odot$ indicates element-wise multiplication, $\sigma(\cdot)$ is the sigmoid function, and $\tanh(\cdot)$ is the hyperbolic tangent function.

After $L$ stacked GGAT layers (we set $L = 3$ in our implementation), the learned protein node representations $\boldsymbol{h}_g^{(L)} \in \mathbb{R}^D$ are aggregated into a disease–pair representation via Eqs. equation 5– equation 7, a process we refer to as Adaptive Pooling. In this step, a two-layer MLP first computes an importance score $s_g$ for each protein $g \in G_p$ in disease pair $p$ (Eq. equation 5). The scores $s_g$ are then softmax-normalized across all proteins in $G_p$ to produce adaptive weights $\alpha_g$ (Eq. equation 6). Finally, the disease–pair representation $P_p$ is obtained as the weighted sum of the corresponding node representations (Eq. equation 7). Adaptive pooling is used because disease-associated protein sets differ in size, and learned weights allow the model to adjust each protein's influence accordingly, unlike fixed pooling rules.

$$s_g = \boldsymbol{w}_2^\top \tanh(\boldsymbol{W}_1 \boldsymbol{h}_g^{(L)}), \tag{5}$$

$$\alpha_g = \frac{\exp(s_g)}{\sum_{n \in G_p} \exp(s_n)}, \tag{6}$$

$$P_p = \sum_{g \in G_p} \alpha_g \boldsymbol{h}_g^{(L)}. \tag{7}$$

Here, $\boldsymbol{W}_1 \in \mathbb{R}^{2D \times D}$ projects protein presentations into a higher-capacity attention space, $\boldsymbol{w}_2^\top \in \mathbb{R}^{1 \times 2D}$ reads out scalar scores.

For prediction, we use a shallow two-layer MLP with ReLU activations to map $P_p$ to a scalar logit, followed by a sigmoid to obtain the predicted comorbidity probability $\hat{y}_p$ for disease pair $p$. The model is trained using the standard binary cross-entropy loss (BCEWithLogitsLoss in PyTorch (Paszke et al., 2019)). The explicit formulas are provided in Appendix B.2 for completeness. To evaluate the contribution of each biological signal, we constructed single-channel GGAT models that use only one type of information at a time, either protein connectivity or disease–protein association information. The connectivity-based variant, GGAT-Connect, employs Node2Vec (Grover & Leskovec, 2016) with parameters $p = q = 1$, where unbiased random walks generate node sequences that are processed by a skip-gram model (Mikolov et al., 2013) to learn protein embeddings. In this unbiased setting, the random walks balance breadth-first and depth-first exploration.

Each random walk consists of gene-encoded protein nodes (e.g., $G_1, G_2, \dots$) encountered along the graph structure. The disease-association variant, GGAT-Disease, constructs protein sets based on disease labels (e.g., Disease A contains all proteins annotated as associated with it) and applies a skip-gram model, similar to Node2Vec, to learn embeddings. These disease labels refer only to disease–protein membership annotations (from the HI benchmark dataset) and do not convey disease–disease relationships or comorbidity outcomes. We refer to this method as Label2Vec. Figure 1 illustrates how these embeddings are derived. To systematically assess predictive power, we train GGAT-Connect and GGAT-Disease separately to make comorbidity predictions.

### 3.2 GGAT Fusion

Next, we develop strategies to integrate information derived from protein connectivity and disease associations for improved comorbidity prediction. We propose two fusion approaches, differing in the integration stage. Recall that the two single-channel models differ only in their initial embeddings: GGAT-Connect uses Node2Vec for connectivity, while GGAT-Disease uses Label2Vec for disease associations.

- **EmbedFusion.** Fusion is applied at the embedding stage by jointly training on both random-walk sequences and disease–protein association sets, yielding integrated embeddings that are processed by a single-channel GGAT.
- **GatedFusion.** Fusion is applied after the single channels by combining the outputs of the GGAT layers from GGAT-Connect and GGAT-Disease through a gated fusion module, which adaptively controls the contribution of each source.

Figure 1 illustrates the EmbedFusion model in comparison to the two single-channel designs. In EmbedFusion, both random-walk sequences and disease–protein sets are provided to a Skip-Gram model, yielding integrated embeddings that capture both connectivity and disease-association information. Importantly, the disease–protein sets contain only membership information (i.e., which proteins are associated with each disease) and do not include any disease–disease or comorbidity labels, ensuring no leakage of the prediction target. These disease–protein associations are taken from the HI benchmark dataset (Menche et al., 2015), which provides curated mappings between diseases and their associated proteins. When only random walks are used, the procedure reduces to Node2Vec and produces connectivity embeddings for the GGAT-Connect channel. When only disease–protein sets are used, it reduces to Label2Vec and produces disease-association embeddings for the GGAT-Disease channel. With both sources jointly provided, the Skip-Gram generates integrated embeddings that are subsequently processed by the GGAT single-channel (Section 3.1) to predict disease comorbidity.

The GatedFusion model (Fig. 2) integrates complementary information from protein connectivity and disease associations through a dual-channel design. For each protein $g$, the model takes as input two node representations derived from the outputs of the corresponding GGAT layers: (i) a connectivity-based representation $\boldsymbol{h}_g^{\mathrm{con}} \in \mathbb{R}^D$ from the GGAT-Connect channel, and (ii) a disease-based representation $\boldsymbol{h}_g^{\mathrm{dis}} \in \mathbb{R}^D$ from the GGAT-Disease channel. We adopt concatenation to combine these representations, which enables the model to capture nonlinear interactions between the two sources while giving the gating mechanism access to both signals when learning their relative weights. The concatenated representations (Eq. equation 8) are then passed through a gated fusion layer (Eqs. equation 9– equation 10) that adaptively balances their contributions.

$$\boldsymbol{x}_g = [\boldsymbol{h}_g^{\mathrm{dis}} \,\|\, \boldsymbol{h}_g^{\mathrm{con}}], \tag{8}$$

$$\gamma_g = \sigma\big(\boldsymbol{W}_2^f \, \rho(\boldsymbol{W}_1^f \boldsymbol{x}_g)\big), \tag{9}$$

$$\tilde{\boldsymbol{h}}_g = (1 - \gamma_g) \odot \boldsymbol{h}_g^{\mathrm{con}} + \gamma_g \odot \boldsymbol{h}_g^{\mathrm{dis}}, \tag{10}$$

Here, the gate $\gamma_g$ controls the relative importance of the two signals, $\boldsymbol{h}_g^{\mathrm{dis}}$ and $\boldsymbol{h}_g^{\mathrm{con}}$, with ReLU $\rho(\cdot)$ extracting informative positive features and sigmoid $\sigma(\cdot)$ normalizing them into soft weights in $[0, 1]$. The fused embedding $\tilde{\boldsymbol{h}}_g$ is then aggregated by the adaptive pooling layer (as in the single-channel design) to form disease-pair representations, which are subsequently passed to the predictor to estimate comorbidity.

## 4 Experiments

### 4.1 Materials

The human interactome (HI) and clinically validated comorbid disease pairs used in this work originate from the dataset described by Velickovic et al. (2017). In that study, relative risk (RR) scores

Table 1: Performance of Disease Comorbidity Prediction: Baselines and GGAT models

| Model | AUROC | AUPRC | Accuracy | MCC |
|---|---|---|---|---|
| GE | $0.5497 \pm 0.0079$ | - | $0.6150 \pm 0.0078$ | - |
| BSE-SVM | $0.6469 \pm 0.0183$ | - | $0.6801 \pm 0.0166$ | - |
| TSPE-NoPE | $0.7971 \pm 0.0146$ | $0.8429 \pm 0.0168$ | $0.7214 \pm 0.0202$ | $0.4340 \pm 0.0299$ |
| TSPE | $0.8009 \pm 0.0152$ | $0.8438 \pm 0.0199$ | $0.7294 \pm 0.0138$ | $0.4578 \pm 0.0378$ |
| GGAT-Connect | $0.8217 \pm 0.0189$ | $0.8595 \pm 0.0186$ | $0.7485 \pm 0.0201$ | $0.4979 \pm 0.0337$ |
| GGAT-Disease | $0.8217 \pm 0.0220$ | $0.8610 \pm 0.0194$ | $0.7476 \pm 0.0230$ | $0.4945 \pm 0.0220$ |
| GGAT-EmbedFusion | $0.8223 \pm 0.0185$ | $0.8599 \pm 0.0165$ | $0.7500 \pm 0.0150$ | $0.4975 \pm 0.0333$ |
| **GGAT-GatedFusion** | **$0.8397 \pm 0.0180$** | **$0.8758 \pm 0.0175$** | **$0.7669 \pm 0.0140$** | **$0.5337 \pm 0.0310$** |

were computed from electronic health records of approximately 13 million patients with one or more diagnoses collected over a four-year period. The curated HI network contains 13,460 protein nodes (identified by their corresponding gene IDs). Following established protocols in previous studies (Velickovic et al., 2017; Akram & Liao, 2019; Qin & Liao, 2025a;b), we constructed a benchmark set of 10,743 disease–disease pairs, labeling pairs with RR > 1 as comorbid (positive) and those with RR ≤ 1 as non-comorbid (negative); this collection is referred to as the RR1 dataset, in which 58.4% of pairs fall into the positive class. The disease–protein associations used to construct Label2Vec and the disease channel are taken directly from the HI benchmark dataset (Menche et al., 2015), which provides curated mappings between each disease and its associated proteins. These annotations contain no comorbidity labels. Additional implementation details, including data splitting protocol, hyperparameter settings, and training procedures, are provided in Appendix C.

## 4.2 COMPARISON OF TRANSFORMER BASELINES AND GGAT MODELS

Table 1 summarizes the disease comorbidity prediction performance of the available baseline models, including GE (Akram & Liao, 2019), BSE (Qin & Liao, 2025a), and TSPE variants (Qin & Liao, 2025b), together with our proposed GGAT models. To the best of our knowledge, these baselines represent all published approaches that operate directly on the human interactome to predict disease–disease relationships. Among them, TSPE remains the strongest existing method. All models are evaluated using four metrics: AUROC, AUPRC, Accuracy, and MCC, reported as the mean and standard deviation across 10-fold cross-validation. Consistent with TSPE, AUROC (Area Under the Receiver Operating Characteristic Curve) is used as the primary metric to ensure comparability with established benchmarks. AUPRC (Area Under the Precision–Recall Curve) complements AUROC by focusing on the positive class, while Accuracy measures overall classification correctness, and MCC (Matthews Correlation Coefficient) (Matthews, 1975) captures balanced performance under label imbalance.

GE (Akram & Liao, 2019) provides a classical geometric embedding of the interactome by preserving approximate geodesic relationships in a reduced-dimensional space, followed by an SVM classifier. BSE (Qin & Liao, 2025a) serves as a supervised dimension-selection framework that identifies the most informative embedding dimensions from any representation, including spectral, geometric, or Node2Vec embeddings. In our experiments, we adopt the strongest variant reported in the BSE study, namely "BSE-SVM", which applies BSE to the Isomap-derived embedding and uses an SVM classifier for prediction. GE and BSE achieve lower performance than TSPE-based and GGAT methods, highlighting the limitations of distance-based and supervised dimension-selection embeddings for this task. For TSPE-based baselines (Qin & Liao, 2025b), we include both TSPE-NoPE and TSPE. TSPE-NoPE uses the Transformer to capture only protein connectivity, whereas TSPE incorporates both connectivity and disease associations. These baselines enable fair evaluation: TSPE-NoPE, the strongest published model that relies solely on protein connectivity, serves as the reference for our GGAT-Connect channel, while TSPE, the overall SOTA baseline, serves as the reference for our GGAT fusion models that integrate connectivity and disease-association information. Since TSPE does not provide a model that uses disease associations alone, no TSPE baseline is directly comparable to our GGAT-Disease channel.

For the single-channel setting, GGAT-Connect improves the primary AUROC from 0.7971 (TSPE-NoPE) to 0.8217, even surpassing the full TSPE model, with consistent and statistically significant gains across all metrics. In particular, the p-values for AUROC improvements over TSPE-NoPE and TSPE are $3.36 \times 10^{-4}$ and $6.60 \times 10^{-4}$, respectively. These results demonstrate that connectivity information alone is sufficient for GGAT single-channel to outperform the SOTA TSPE. Similarly, GGAT-Disease attains performance comparable to GGAT-Connect and significantly exceeds both

Table 2: Gated and Non-Gated Model Variants on the Connect Channel

| Model | AUROC | AUPRC | Accuracy | MCC |
|---|---|---|---|---|
| **Gated Variants** | | | | |
| **GGAT** | **0.8217 ± 0.0189** | **0.8595 ± 0.0186** | **0.7485 ± 0.0201** | **0.4979 ± 0.0337** |
| GGCN | 0.8176 ± 0.0162 | 0.8548 ± 0.0146 | 0.7469 ± 0.0156 | 0.4885 ± 0.0301 |
| GGraphSAGE | 0.8085 ± 0.0231 | 0.8524 ± 0.0238 | 0.7374 ± 0.0247 | 0.4750 ± 0.0443 |
| **Non-Gated Variants** | | | | |
| GAT | 0.5504 ± 0.0387 | 0.6226 ± 0.0261 | 0.5685 ± 0.0461 | 0.1115 ± 0.0635 |
| GCN | 0.7828 ± 0.0202 | 0.8378 ± 0.0200 | 0.7138 ± 0.0190 | 0.4329 ± 0.0300 |
| GraphSAGE | 0.8061 ± 0.0255 | 0.8553 ± 0.0233 | 0.7280 ± 0.0199 | 0.4617 ± 0.0445 |

Table 3: Gated and Non-Gated Model Variants on the Disease Channel

| Model | AUROC | AUPRC | Accuracy | MCC |
|---|---|---|---|---|
| **Gated Variants** | | | | |
| **GGAT** | **0.8217 ± 0.0219** | **0.8610 ± 0.0194** | **0.7476 ± 0.0229** | **0.4945 ± 0.0366** |
| GGCN | 0.8072 ± 0.0175 | 0.8530 ± 0.0134 | 0.7311 ± 0.0183 | 0.4654 ± 0.0309 |
| GGraphSAGE | 0.8105 ± 0.0249 | 0.8564 ± 0.0220 | 0.7374 ± 0.0217 | 0.4759 ± 0.0381 |
| **Non-Gated Variants** | | | | |
| GAT | 0.5685 ± 0.0421 | 0.5796 ± 0.1825 | 0.5696 ± 0.0241 | 0.1253 ± 0.0593 |
| GCN | 0.7715 ± 0.0172 | 0.8185 ± 0.0168 | 0.7040 ± 0.0158 | 0.4100 ± 0.0247 |
| GraphSAGE | 0.8169 ± 0.0224 | 0.8601 ± 0.0216 | 0.7390 ± 0.0158 | 0.4817 ± 0.0396 |

TSPE baselines, demonstrating that disease association information alone can also be effectively captured by our single-channel architecture. For the fusion setting, GGAT-EmbedFusion achieves a slight increase in AUROC and MCC over the single-channel variants but fails to improve the other metrics consistently. In contrast, GGAT-GatedFusion achieves the best overall performance, with consistent and statistically significant improvements over both GGAT single-channel models and the TSPE baselines. In particular, the p-values for AUROC gains over GGAT-Connect and GGAT-Disease are $8.24 \times 10^{-5}$ and $2.61 \times 10^{-4}$, respectively. Moreover, GGAT-GatedFusion improves AUROC by 3.88 percentage points over the SOTA TSPE baseline ($p = 2.61 \times 10^{-4}$), which already integrates protein connectivity and disease associations, underscoring the effectiveness of gated fusion in integrating complementary biological information. Overall, these results establish GGAT-GatedFusion as a new SOTA for comorbidity prediction, surpassing the Transformer-based TSPE baseline by a substantial margin. The improvements reflect our key design contributions: leveraging local graph attention for more biologically meaningful aggregation, introducing a gating mechanism to adaptively regulate information flow, and enabling flexible multichannel fusion to integrate heterogeneous biological signals.

### 4.3 ABLATION STUDY

Table 2 and Table 3 compare gated and non-gated GNN variants using the same embeddings as in the GGAT-Connect and GGAT-Disease channels. These architectures differ primarily in how they aggregate neighborhood information. GAT (Velickovic et al., 2017) employs multi-head attention to learn edge-specific coefficients and aggregates neighbors, including self-loops, through an attention-weighted sum. GCN (Kipf, 2016) replaces attention with degree-normalized averaging over neighbors, also including self-loops and without learnable edge weights. GraphSAGE (Hamilton et al., 2017) separates self and neighbor information by first aggregating neighbors without self-loops and then concatenating the node's own features with the aggregated message, followed by a linear transformation and a ReLU activation. GGAT, GGCN, and GGraphSAGE represent the gated variants of these models, where a GRU module is applied after each layer to regulate the feature updates.

GGAT achieves the highest overall performance across both the Connect and Disease model variants, as its attention mechanism selectively up-weights biologically meaningful neighbors, while the GRU gate stabilizes deep attention propagation and prevents attention collapse. In contrast, without gating, vanilla GAT shows unstable optimization dynamics on noisy interactome and suffers from severe performance degradation. In GCN, self features are mixed with all neighbors through degree-normalized averaging, which makes the updates sensitive to degree variation and neighborhood noise. The GRU gate helps stabilize these updates by suppressing overly aggressive or unreliable changes, leading to a clear improvement in GGCN over GCN. In contrast, GraphSAGE keeps self and neighbor information separate by first aggregating neighbors without self-loops and then

fusing them through a learnable linear layer. This produces more stable updates, so the GRU gate provides only marginal improvements in the connect channel (Table 2) and does not consistently help on the disease channel, where GGraphSAGE even performs worse than GraphSAGE (Table 3). This indicates that gating is less beneficial for GraphSAGE than for GAT or GCN under our task setting. Although both GCN and GraphSAGE are generally more stable than GAT in the single-channel setting, they rely on uniform neighbor averaging and therefore cannot learn edge-specific importance. As a result, their gated variants, GGCN and GGraphSAGE, still remain consistently below GGAT, which benefits simultaneously from adaptive attention and gated stabilization.

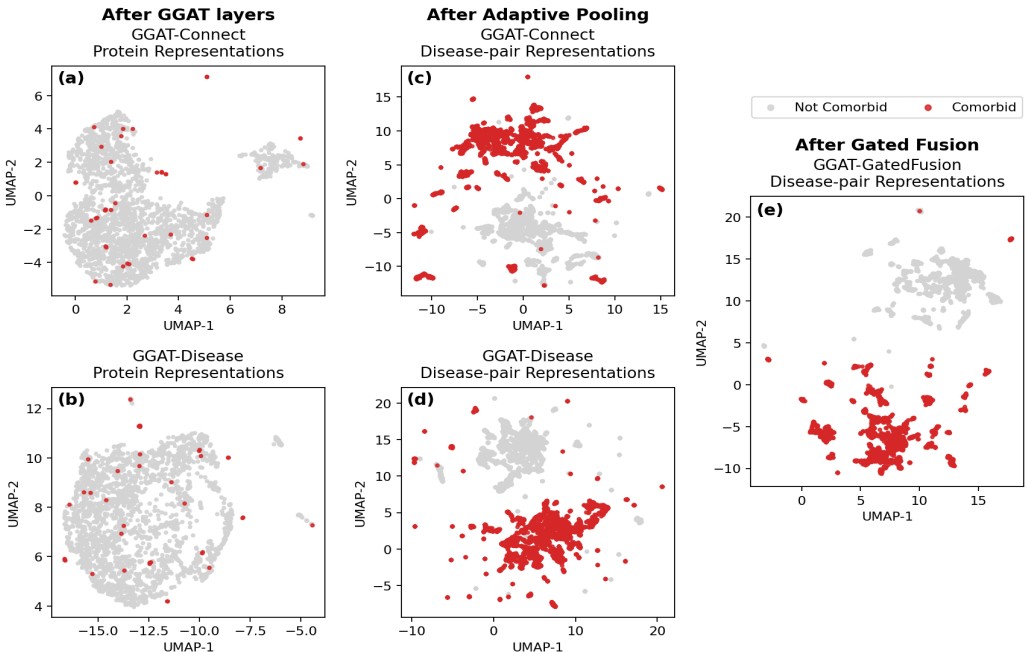

Figure 3: UMAP projections of learned features at different stages of the GGAT framework. (a,b) Protein-level features from the connectivity and disease channels after GGAT layers. (c,d) Disease–pair features after adaptive pooling, where informative nodes are emphasized to form more structured representations. (e) Fused disease–pair features from the gated fusion model, which integrates complementary signals from the two channels. While GGAT layers capture meaningful local node features, the separation between comorbid (red) and non-comorbid (gray) pairs becomes progressively clearer after pooling and is further enhanced by gated fusion.

## 4.4 INTERPRETABILITY

We next examined the interpretability of the proposed GGAT framework using UMAP projections (McInnes et al., 2018). Because comorbid protein nodes are sparse, the visualizations are generated from the entire dataset. At the protein node level (Figs. 3a,b), the representations learned up to the GGAT layers alone do not yield a clear separation between comorbid and non-comorbid related nodes. This is expected: individual proteins often participate in multiple diseases, and different disease pairs may exhibit distinct comorbidity outcomes, so comorbidity cannot be inferred from single-node features alone. Nevertheless, the GGAT layers play a critical role at this stage by encoding localized neighborhood signals within the interactome, while the gating mechanism regulates how newly aggregated information is combined with prior context to reduce noise and oversmoothing. These enriched features form the necessary foundation that enables adaptive pooling to meaningfully aggregate node-level information into higher-level disease–pair representations.

After adaptive pooling (Figs. 3c,d), the separation between comorbid and non-comorbid disease pairs becomes more apparent. By weighting informative proteins within each disease module, adaptive pooling transforms noisy node-level features into structured disease–pair representations that more faithfully capture comorbidity relationships. Finally, the GatedFusion model (Fig. 3e) sharpens this separation further by integrating complementary signals from the GGAT-Connect channel and the GGAT-Disease channel, yielding a more interpretable feature space in which comorbid pairs form distinct clusters in UMAP. This enhanced clustering indicates that the fusion layer does more

than simply combine signals, as it adaptively balances their contributions in a way that highlights their joint relevance to comorbidity.

The progression observed in the UMAP visualizations, from node-level to pooled disease–pair features and ultimately to fused representations, provides a coherent narrative for our key contributions: local attention preserves biologically meaningful neighbor information, gating stabilizes feature propagation and emphasizes context, adaptive pooling constructs structured disease–pair features from heterogeneous nodes, and multichannel fusion integrates complementary biological evidence into a unified, more interpretable space.

## 5 CONCLUSION

We introduced the Gated Graph Attention Network (GGAT) framework for learning from protein–protein interaction networks (human interactome) to predict disease comorbidity. The limitations of the SOTA TSPE method stem from its reliance on global pairwise attention, which neglects graph topology, and its use of simple feature concatenation to combine connectivity and disease associations, which is both inefficient and inflexible. GGAT addresses these limitations through local neighborhood attention, gating, and gated fusion. Experiments on the benchmark dataset show significant and consistent improvements of more than 3 percentage points over the SOTA TSPE method across all evaluation metrics, with the GatedFusion variant achieving the best overall performance. The multichannel design in GatedFusion offers a flexible way to incorporate diverse biological signals while mitigating the dimensionality and scalability issues of concatenation-based positional encodings in TSPE. Our interpretability analysis demonstrates that while GGAT layers capture protein-level features, the separation between comorbid and non-comorbid disease pairs becomes more apparent after adaptive pooling and is further enhanced by gated fusion, resulting in progressively more interpretable feature spaces. Taken together, these results establish GGAT as a new SOTA for comorbidity prediction. Given the GGAT framework's inherent suitability for modeling relationships between labeled node groups, the framework can be readily adapted and applied to other graph problems that involve group-to-group prediction.

**Limitations.** A current limitation of this work is that evaluation is restricted to the HI interactome and the RR dataset, which remains the only established benchmark for interactome-based comorbidity prediction. This constraint limits our ability to systematically assess generalizability across alternative interactomes. In addition, protein–protein interaction networks are incomplete, undirected, and not context-specific, meaning they cannot represent condition-dependent or directional biological relationships. These limitations are shared by all existing interactome-based comorbidity models. Nevertheless, prior work indicates that incomplete interactomes retain enough structural signal to support network-based disease–disease modeling. Moreover, the comorbidity labels used in this benchmark are derived from relative-risk estimates, which are known to be imperfect and influenced by demographic and clinical factors; however, RR remains the established ground-truth definition for interactome-based comorbidity prediction, and all prior frameworks rely on the same labels. Another limitation concerns the scope of our fusion module, whose flexibility applies only to node-aligned learned representations that map one-to-one onto protein nodes. It is not a general multimodal integration framework, and the present study uses only the two interactome-derived sources available in this benchmark.

**Future work.** Future work will explore three directions. First, we plan to extend GGAT to larger and more up-to-date human interactome resources, such as HuRI (Luck et al., 2020), BioPlex (Huttlin et al., 2021), and STRING (Szklarczyk et al., 2023), to uncover previously unrecognized disease comorbidities and protein-level mechanisms. Second, the multichannel fusion architecture enables incorporation of additional biological modalities, such as sequence-derived protein embeddings, e.g., ESM (Lin et al., 2023), or biomedical text embeddings, e.g., BioBERT (Lee et al., 2020), to assess whether finer-grained biochemical or semantic signals can further strengthen disease-module–level relationship prediction. Third, we plan to investigate the integration of heterogeneous biological and clinical modalities, including pathway-level information, regulatory signals, phenotypic data, and longitudinal EHR-derived modalities. Integrating these multimodal sources is a more complex but valuable direction that will require dedicated methodological development.

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

## A   EQUIVALENCE OF THE CONCATENATION FORM AND THE SPLIT SUMMATION FORM FOR GRU GATES

**Claim.** For any layer $l$, node $i$, hidden state $\boldsymbol{h}_i^l \in \mathbb{R}^H$ and input $\boldsymbol{x}_i^l \in \mathbb{R}^F$, the gate computed by the concatenation form

$$\boldsymbol{g} = \sigma\left(\boldsymbol{W}^l[\boldsymbol{h}_i^l \,\|\, \boldsymbol{x}_i^l] + \boldsymbol{b}^l\right), \quad \boldsymbol{W}^l \in \mathbb{R}^{D \times (H+F)}, \ \boldsymbol{b}^l \in \mathbb{R}^D, \tag{A.1}$$

is identical to the gate computed by the split summation form

$$\boldsymbol{g} = \sigma\left(\boldsymbol{U}^l \boldsymbol{h}_i^l + \boldsymbol{V}^l \boldsymbol{x}_i^l + \boldsymbol{b}^l\right), \quad \boldsymbol{U}^l \in \mathbb{R}^{D \times H}, \ \boldsymbol{V}^l \in \mathbb{R}^{D \times F}, \ \boldsymbol{b}^l \in \mathbb{R}^D, \tag{A.2}$$

where $\sigma(\cdot)$ is any element-wise nonlinearity (e.g., sigmoid). The two parameterizations in (A.1) and (A.2) define the same function class.

**Proof.** Write the concatenated vector as

$$\boldsymbol{y}_i^l = \begin{bmatrix} \boldsymbol{h}_i^l \\ \boldsymbol{x}_i^l \end{bmatrix} \in \mathbb{R}^{H+F}. \tag{A.3}$$

Partition $\boldsymbol{W}^l$ column-wise into two blocks,

$$\boldsymbol{W}^l = [\boldsymbol{U}^l \ \boldsymbol{V}^l], \quad \boldsymbol{U}^l \in \mathbb{R}^{D \times H}, \ \boldsymbol{V}^l \in \mathbb{R}^{D \times F}. \tag{A.4}$$

By block matrix multiplication,

$$\boldsymbol{W}^l \boldsymbol{y}_i^l = [\boldsymbol{U}^l \ \boldsymbol{V}^l] \begin{bmatrix} \boldsymbol{h}_i^l \\ \boldsymbol{x}_i^l \end{bmatrix} = \boldsymbol{U}^l \boldsymbol{h}_i^l + \boldsymbol{V}^l \boldsymbol{x}_i^l. \tag{A.5}$$

Plugging (A.5) into (A.1) yields (A.2). Conversely, given any $\boldsymbol{U}^l, \boldsymbol{V}^l$ we can define $\boldsymbol{W}^l = [\boldsymbol{U}^l \ \boldsymbol{V}^l]$, so that (A.1) recovers (A.2). The bias $\boldsymbol{b}^l$ is shared in both (A.1) and (A.2). Hence the two expressions (A.1) and (A.2) are algebraically equivalent for all $(\boldsymbol{h}_i^l, \boldsymbol{x}_i^l)$.

# B   ADDITIONAL IMPLEMENTATION DETAILS

## B.1   STANDARD MULTI-HEAD GAT FORMULATION

For completeness, we briefly summarize the standard multi-head GAT updates (Velickovic et al., 2017) used inside each GGAT layer. For the $k$-th head at layer $l$, the unnormalized attention score $e_{ij}^{l,(k)}$ between node $i$ and a neighbor $j \in \mathcal{N}(i)$ is

$$e_{ij}^{l,(k)} = \text{LeakyReLU}\Big( (\boldsymbol{a}^{l,(k)})^\top \big[ \boldsymbol{W}^{l,(k)} \boldsymbol{h}_i^l \,\|\, \boldsymbol{W}^{l,(k)} \boldsymbol{h}_j^l \big] \Big), \tag{11}$$

$$\alpha_{ij}^{l,(k)} = \frac{\exp\big(e_{ij}^{l,(k)}\big)}{\sum_{m \in \mathcal{N}(i)} \exp\big(e_{im}^{l,(k)}\big)}, \tag{12}$$

$$\boldsymbol{u}_i^{l,(k)} = \text{ELU}\Big( \sum_{j \in \mathcal{N}(i)} \alpha_{ij}^{l,(k)} \boldsymbol{h}_j^l \Big), \tag{13}$$

$$\boldsymbol{x}_i^l = \big\|_{k=1}^{K} \boldsymbol{u}_i^{l,(k)}. \tag{14}$$

Here, $\boldsymbol{W}^{l,(k)} \in \mathbb{R}^{F' \times F}$ is the head-specific projection matrix and $\|$ denotes concatenation.

For the $k$-th head, node features $\boldsymbol{h}_i^{l,(k)}, \boldsymbol{h}_j^{l,(k)} \in \mathbb{R}^{F'}$ represent the projected embeddings of node $i$ and its neighbor $j$, and $\mathcal{N}(i)$ denotes the neighborhood of node $i$. The unnormalized attention score $e_{ij}^{l,(k)}$ for edge $(i,j)$ is computed by applying a learnable vector $\boldsymbol{a}^{l,(k)} \in \mathbb{R}^{2F'}$ to the concatenated projected features of nodes $i$ and $j$ (Eq. equation 11). The score $e_{ij}^{l,(k)}$ is then normalized across all neighbors to produce attention weights $\alpha_{ij}^{l,(k)}$ (Eq. equation 12). The weighted neighbor features are aggregated and passed through an ELU activation to yield the head-specific representation (Eq. equation 13), which are subsequently concatenated across heads (Eq. equation 14).

## B.2   PREDICTION LAYER AND LOSS

Given a disease–pair representation $P_p$, the predictor is a two-layer MLP with ReLU activation,

$$\ell_p = \boldsymbol{w}_4^\top \rho(\boldsymbol{W}_3 \rho(P_p)), \tag{15}$$

$$\hat{y}_p = \sigma(\ell_p), \tag{16}$$

where $\rho(\cdot)$ is ReLU and $\sigma(\cdot)$ is the sigmoid. The model is trained with the standard binary cross-entropy loss,

$$\mathcal{L}_p = -y_p \log \sigma(\ell_p) - (1 - y_p) \log\big(1 - \sigma(\ell_p)\big), \tag{17}$$

where $y_p \in \{0, 1\}$ is the ground-truth comorbidity label.

Table 4: Training and architecture hyperparameters for the GGAT models used in this study.

| Component | Setting |
|---|---|
| Epochs | 3000 |
| Loss function | BCEWithLogitsLoss |
| **GGAT backbone** | |
| Number of GGAT layers | 3 |
| Hidden dimension per head | 8 |
| Heads (layer 1) | 8 |
| Heads (layer 2) | 4 |
| Heads (layer 3) | 1 |
| Intermediate node dim | 32 |
| GAT dropout | 0.4 |
| **Gating and pooling** | |
| Pooling hidden dim | 64 |
| **RR predictor** | |
| Predictor input dim | 32 |
| Predictor hidden dim | 32 |
| Output dim | 1 |

## C   IMPLEMENTATION DETAILS

**Data splitting protocol.**   We perform stratified 10-fold cross-validation on the 10,743 disease–disease pairs, preserving the positive/negative disease comorbidity pair ratio in each fold. For each run, 9 folds are used for training, and one for testing. The test fold is never used during model selection or early stopping.

**Training procedure and model hyperparameters.**   We train all models with the Adam optimizer (learning rate 0.005, weight decay $5 \times 10^{-4}$). Single-channel GGAT models run for 3000 epochs, while the GatedFusion variant runs for 2000 epochs. All hyperparameters are summarized in Table 4.

**Implementation environment.**   We run all experiments in the Google Colab environment on an NVIDIA Tesla T4 GPU with CUDA 12 support.

## LLM USAGE

We used large language models (LLMs), such as ChatGPT, only for grammar checking and language polishing.

