# OpenReview forum: "Gated Graph Attention Networks with Multichannel Fusion for Disease Comorbidity Prediction"
_ICLR.cc/2026/Conference — ICLR 2026 Conference Withdrawn Submission_

### Official Review · Reviewer_GEJD · 2025-10-29

**Soundness:** 3
**Presentation:** 2
**Contribution:** 2
**Rating:** 4
**Confidence:** 5

**Summary:**

The paper presents a gated graph attention network (GGAT)  that predicts disease comorbidity using human protein-protein interaction graphs. GGAT utilize GAT layers to define attention based neighbor information aggragation. To avoid over smoothing, they insert GRU style gates between the GAT layers.These gates control how much new neighbor information to accept and how much of previous information to keep. Each protein has 2 types of initial embeddings, one connectivity embedding resulting from Node2vec with randomwalk on PPI network and one disease-association embedding resulting from Label2vec. The model combines these 2 embeddings with 2 different fusion model:EmbedFusion for early fusion, where it merges them into one embedding then they run GGAT or GatedFusion for late fusion, where it runs a GGAT for each type (in parallel)  then use a learned gate to fuse them into one embedding. To predict the comorbidity for a disease pair, the model gets the proteins linked to each one of the diseases and produce a disease-pair embedding using adaptive pooling, which is passed to a MLP classifier to output the probability of the 2 diseases being comorbid. The experiments showed that the GatedFusion based GGAT has the best results overall, compared to the Transformer (TSPE) baseline

**Strengths:**

- the usage of GRU gates between GAT layers to regulate information flow to see how much new neighbor information to accept and how much of the previous representation to keep is an interesting idea.
 - they introduce the GatedFusion module that combines connectivity and disease-association embeddings for proteins with a learned gate
-leveraging protein–protein interaction networks for this Disease comorbidity prediction is well-motivated.

**Weaknesses:**

-Constribution is very limited as combining GAT and GRU, both the gating and fusion components are well-established concepts in prior graph learning works (e.g., GatedGCN, GCN-GRU).
-The claimed novelty of “multichannel fusion” is essentially feature concatenation followed by gating, which offers minimal conceptual advancement over prior multimodal GNN frameworks.
- The state of the art used for comparison only focus on TSPE which may not be enough to validate the model. It is not including gated networks or GAT models results.
- There is no clear theoretical analysis or ablation to demonstrate why gating specifically benefits comorbidity prediction beyond empirical results.
-Only one baseline (TSPE) is compared; missing comparisons with GCN, GatedGCN, and other biomedical GNN models limits the strength of the evaluation.
Ablations and hyperparameters study missing. There is no comparison with or without GRU gates, the different pooling types (why adaptive pooling in particular). It helps us see which parts matter most.
- Paper writing should be improved. disease-association information is mentioned but not explained. How they obtained.
- including disease-association information information as the initial embeding may cause label leakege.

**Questions:**

Q1. Did you check if there is any information leakage when splitting into training and testing? Does a disease appear in both training and testing?
Q2. Are  Node2Vec and Label2Vec computed inside each training fold?
Q3. Could be usufull to Include an ablation study section where you show what happens if you remove GRU, if you replace adaptive pooling with mean or another type.
Q4. Could you please explain why GatedFusion is better than EmbedFusion
Q5. Did you try Node2Vec with different parameters values (p,q)? You can include a sensitivity study about it. Same thing applies to Label2Vec
Q6. You mentioned that TSPE has a concatenation cost, can you compare it with GGAT ?
Q7. Could you clarify the role of the Label2Vec embeddings? How are these embeddings trained — jointly with GGAT or pre-trained and fixed? How are diseases represented in the embedding space (e.g., one embedding per protein, per disease, or per disease pair)?

---

> ### Author Response · Authors · 2025-11-21
> **Response to Reviewer GEJD (1)**
>
> We sincerely thank the reviewer for the detailed and thoughtful feedback, which has helped us clarify key aspects of the architectural design and improve multiple parts of the manuscript. We respond to each comment in detail below.
>
> **Response to weakness #1:**
>
> We would like to clarify that our manuscript is application-driven: our aim is not to introduce a new GNN primitive, but to design a system capable of modeling disease-conditioned protein representations and constructing set-level disease pair embeddings. While GAT layers and GRU-style gating are individually well-known, to the best of our knowledge no prior work combines (i) node-level local attention, (ii) layer-wise GRU gating for stabilizing information propagation, (iii) embed-level fusion, and (iv) multi-channel biological information fusion in this domain.
>
> Regarding the reviewer’s mention of GatedGCN and GCN–GRU models: to our knowledge, GCN+GRU combinations have not been used in disease comorbidity prediction. GatedGCN (Bresson & Laurent, 2017) performs edge-level message gating, modulating each neighbor message before aggregation, whereas GCN–GRU models such as Zhao et al. (2018) use GRUs for temporal sequence modeling, capturing dynamics across time. In contrast, our GRU gate operates at the node-state level across GAT layers, regulating how much new attention-aggregated information to incorporate versus how much prior representation to retain. While similar gating ideas to our GGAT may exist in unrelated domains, we have not found any application of such gating in GCN and GRU combinations.
>
> Our architectural motivation comes directly from the limitations of the prior state-of-the-art TSPE model, which demonstrate the utility of attention but its global attention mechanism is less suited for capturing local interactome structure. We therefore adopt GAT, which provides localized and edge-aware attention tailored to PPI neighborhoods. However, because the interactome is noisy, attention alone cannot ensure stable multi-layer propagation. To address this, we incorporate a layer-wise GRU gating mechanism to stabilize information flow across GAT layers. Our design thus begins from GAT—motivated by the domain’s SOTA method—and extends it with task-specific gating to meet the requirements of comorbidity prediction.
>
> Also, our gating mechanism is architecturally novel in the comorbidity prediction setting: it is not a temporal GRU widely used in prior GNN or sequence models, but a GRU-style, layer-wise fusion gate specifically adapted to regulate attention propagation on noisy biological graphs.
>
> On the fusion side, we explored two task-motivated designs: (1) Embedding-Level EmbedFusion, which introduces a mixed random-walk–plus–disease-set-to-vec embedding, (2) Channel-wise GatedFusion, which adaptively combines the outputs of single GGAT channels after independent processing. Both are not explored in prior work. Although EmbedFusion underperforms GatedFusion, the experiments reveal an informative pattern: early fusion introduces more information but also more noise, whereas late gated fusion is more effective at integrating heterogeneous biological signals. We believe this observation is not only relevant to comorbidity prediction but also potentially valuable to other multimodal biological graph learning settings.
>
> [1] Bresson, Xavier, and Thomas Laurent. "Residual gated graph convnets." arXiv preprint arXiv:1711.07553 (2017).
>
> [2] Xujiang Zhao, Feng Chen, Jin-Hee Cho [Deep Learning for Predicting Dynamic Uncertain Opinions in Network Data], Bigdata (2018)
>
> **Response to weakness bullet point 1:**
>
> We appreciate this valuable suggestion. In the revised submission, we have included a full set of gated and non-gated GNN baselines built on identical inputs. Specifically, we compare Gated GAT, vanilla GAT, Gated GraphSAGE, GraphSAGE, Gated GCN, and GCN under both the connectivity and disease channels (Tables 2 & 3).
>
> These additional evaluations provide a comprehensive assessment of the architectural choices. Across all these controlled comparisons, GGAT consistently achieves the strongest overall performance, demonstrating that both local attention and layer-wise gating contribute to the effectiveness of our model beyond the TSPE baseline.

---

> ### Author Response · Authors · 2025-11-21
> **Response to Reviewer GEJD (2)**
>
> **Response to weakness bullet point 2:**
>
> We appreciate this suggestion. In the revised submission, we now provide a substantially more comprehensive evaluation through four levels of comparison:
>
> (a) Recent comorbidity-focused baselines: GE, BSE, TSPE — covering all published interactome-based comorbidity prediction models to date (Table 1).
> (b) Standard GNN baselines: GCN, GraphSAGE, and vanilla GAT evaluated under identical input embeddings (Tables 2 & 3).
> (c) Gated vs. non-gated variants: Gated and non-gated versions of the same GNN backbones (GAT, GraphSAGE, GCN) under the same controlled setting (Tables 2 & 3), directly isolating the contribution of GRU gating.
> (d) Channel-level ablations: Connectivity-only, disease-only, and fusion variants (Table 1), showing how different biological evidence contributes to performance.
>
> All models use the same pooling strategy to ensure fairness. We adopt adaptive pooling because disease-associated protein sets vary in size, and adaptive pooling naturally aggregates such variable-sized sets by weighting each protein’s contribution to the disease-level representation. In contrast, fixed operations such as max or mean pooling do not account for set-size variability and may oversmooth or discard informative signals. We have added an explanation of this design choice to the revised manuscript.
>
> Across these evaluations, GGAT models consistently achieve the strongest overall performance, demonstrating that both local attention and layer-wise gating meaningfully benefit comorbidity prediction beyond empirical improvement over TSPE. We have additionally included the hyperparameter settings in the appendix for completeness.
>
> **Response to weakness bullet point 3:**
>
> Thank you for the comment. The manuscript already describes that the disease–association information comes from the disease–protein sets used as inputs to the Skip-Gram model in EmbedFusion (Fig. 1). To avoid any confusion, we have revised the text to clearly state that these disease–protein sets are obtained from the HI benchmark dataset (Menche et al., Science 2015), which provides curated mappings between diseases and their associated proteins. The Methods section has been updated to make this explanation clearer.
>
> [1] Menche, Jörg, Amitabh Sharma, Maksim Kitsak, Susan Dina Ghiassian, Marc Vidal, Joseph Loscalzo, and Albert-László Barabási. "Uncovering disease-disease relationships through the incomplete interactome." Science 347, no. 6224 (2015): 1257601.
>
> **Response to weakness bullet point 4:**
>
> We would like to clarify that no label leakage occurs in our framework. The task is to use (i) the protein–protein interaction graph (the human interactome) and (ii) disease–protein association information (disease labels) to predict the comorbidity label of a disease pair.
>
> The disease labels used at the node level simply indicate which diseases each protein is associated with; they do not contain any information about disease–disease relationships or comorbidity outcomes. The comorbidity labels (RR > 1 vs. RR ≤ 1) used for classification are never provided to the protein node embeddings.
>
> The disease-association channel uses only membership information (whether a protein belongs to a given disease gene set), which is the same information used in prior baselines such as TSPE and is not derived from comorbidity outcomes. This information is also legitimately available at inference time, because disease gene sets are part of the problem definition: to evaluate a disease pair (A, B), their associated protein sets must be known in order to (1) identify the relevant protein nodes, (2) aggregate their learned embeddings via adaptive pooling, and (3) form the disease-pair representation for prediction. For example, if disease A is associated with proteins {p1, p2, p3} and disease B with {p3, p4, p5}, these membership sets are required to construct the pair (A, B) and do not reveal the comorbidity label.
>
> The prediction task is defined on disease pairs, not on individual diseases. Thus, the training and test sets contain only disease–pair representations, and no disease pair ever appears in both sets.
> Thus, the model does not use any information that would be unavailable or illegal at prediction time, nor does it leak comorbidity labels into the learning process. To avoid any confusion, we have updated the manuscript to make this point explicit.

---

> ### Author Response · Authors · 2025-11-21
> **Response to Reviewer GEJD (3)**
>
> **Response to Q1:**
>
> Thank you for the question. This point is addressed in our response to the weakness section. Briefly, our prediction task is defined on disease pairs, the split is performed at the pair level, and node-level disease annotations contain no comorbidity information, so no leakage occurs.
>
> **Response to Q2:**
>
> Node2Vec is computed once on the full HI interactome, and Label2Vec is computed once from the disease–protein membership annotations in the benchmark dataset. Because the cross-validation split is applied only at the disease-pair level (not at the protein or graph level), there is no need to recompute node embeddings inside each fold. The training and test disease pairs are constructed from different protein sets, so the input disease pair representations for training and testing differ accordingly. This follows the standard practice used in GE, BSE, and TSPE.
>
> **Response to Q3:**
>
> We thank the reviewer for the suggestion. The revised manuscript now includes ablations on GRU gating and pooling strategies; these results and the rationale for choosing adaptive pooling are summarized in our response to the Weakness section.
>
> **Response to Q4:**
>
> Thank you for the question. This explanation is provided in our first response to the Weakness section regarding “fusion design.” Briefly, early fusion (EmbedFusion) mixes heterogeneous signals before any structure-aware processing, which introduces both information and noise. In contrast, late gated fusion (GatedFusion) combines the already-learned representations from the separate GGAT channels and integrates them through a gating mechanism, producing more stable and effective embeddings. This accounts for the stronger performance of GatedFusion.
>
> **Response to Q5:**
>
> Node2Vec in our work follows exactly the same configuration as in the TSPE baseline in order to ensure a fair comparison, since our contribution focuses on the GGAT architecture and fusion design rather than on optimizing random-walk hyperparameters. Performing a sensitivity analysis on p and q would change the baseline setting and confound comparability with prior work.
>
> We also clarify that p and q apply only to Node2Vec’s random-walk generation. Label2Vec does not use random walks. Its embeddings are learned from disease–protein sets via Skip-Gram, so these parameters are not applicable.
>
> **Response to Q6:**
>
> The two models are based on fundamentally different design principles, and these architectural differences already explain the advantage of GGAT over TSPE.
>
> In TSPE, the subgraph positional encoding incorporates disease information by concatenating a projected Graph Encoder Embedding (GPE) with the connectivity and Laplacian positional features. In the original TSPE implementation, the GPE projection was limited to 8 dimensions; larger dimensions were avoided because increasing d_"GPE"  inflates the computational cost of each Transformer self-attention and encoder–decoder attention block.
> In contrast, GGAT does not merge all information into one enlarged embedding. Instead, it processes connectivity and disease signals in separate GGAT channels with moderate dimensionality and fuses them through a lightweight gated module. This design avoids expanding the node embedding dimension and prevents the cost growth associated with the TSPE’s concatenation-based design.
>
> We have added a clearer explanation of these differences to the revised manuscript.
>
> **Response to Q7:**
>
> To avoid ambiguity, we use input embeddings to refer to fixed, pre-trained node features (Label2Vec), and learned representations to refer to the latent vectors produced by GGAT during training.
>
> Label2Vec provides a disease–association input embedding for each protein node. These vectors are pre-trained once using Skip-Gram on disease–protein sets (treated as unordered “disease sentences”) and used as fixed input node features to GGAT; they are not updated during GGAT training.
>
> During training, GGAT learns new protein representations through graph attention and layer-wise gating. Each protein obtains a learned representation vector, diseases are represented implicitly through the sets of their associated protein vectors, and a disease pair is represented by adaptively pooling the learned vectors from the two corresponding protein sets. The same representation scheme is used at test time: proteins retain their learned representation vectors, and diseases and disease pairs are constructed in the same way from their associated protein sets.

---

### Official Review · Reviewer_Xd2g · 2025-10-30

**Soundness:** 2
**Presentation:** 2
**Contribution:** 1
**Rating:** 2
**Confidence:** 4

**Summary:**

This paper studies disease comorbidity prediction on the human interactome using a Gated Graph Attention Network (GGAT) that couples local GAT message passing with GRU-style gating and introduces two fusion strategies (EmbedFusion and GatedFusion) to combine connectivity- and disease-association–derived embeddings

**Strengths:**

- Clear decomposition of signals (connectivity vs. disease association) and a simple, reproducible fusion design (single-channel vs. dual-channel) make the study easy to follow and reimplement.
- Consistent improvements across four metrics were observed.

**Weaknesses:**

- Much of the model is built from well-established components (GAT layers, GRU gating, etc.), so the methodological novelty appears limited.

- Since GAT and GRU are well known, enumerating nearly all equations is unnecessary; streamlining the math would improve focus on the new contributions.

- Protein connectivity and disease associations could be complemented by pretrained single-entity embeddings (e.g., ESM for proteins, BioBERT for disease text) to enrich representations.

- Experiments rely on a single data source with a general evaluation; deeper analyses of generalizability (e.g., across alternative interactomes or disease subsets) are missing.

- Important experimental details are insufficiently specified (e.g., data splitting protocol, hyperparameter selection/search), making it hard to assess rigor.

- The manuscript does not discuss any limitations.

**Questions:**

None

---

> ### Author Response · Authors · 2025-11-21
> **Response to Reviewer Xd2g (1)**
>
> We sincerely thank the reviewer for the thoughtful feedback, which has helped us improve the manuscript and expand the experimental evaluation. We provide detailed clarifications and updates in response to each comment below.
>
> **Response to weakness #1:**
>
> We would like to clarify that our manuscript is application-driven rather than focused on introducing new primitives. We agree that GAT layers and GRU-style gating are well-established components.
>
> However, the goal of this work is to design an architecture that advances the state of the art by better modeling disease-specific protein sets and disease–disease relationships, which is challenging and fundamentally different from standard node-level GNN tasks.
>
> Importantly, our gating mechanism is architecturally novel in the comorbidity prediction setting: it is not a temporal GRU but a GRU-style, layer-wise fusion gate specifically adapted to regulate attention propagation on noisy biological graphs.
>
> Therefore, our novelty lies in the following components:
>
> •	A fundamentally different architectural design from prior interactome-based comorbidity prediction methods, explicitly modeling disease-specific protein sets and disease–disease interactions rather than relying on node-level GNN embeddings, and advancing the state of the art in prediction.
>
> •	Local biological attention via GAT. To address the limitation of SOTA method TSPE, which relies solely on global pairwise attention, we leverage graph attention over local PPI neighborhoods to capture biologically meaningful and topologically coherent interactions within the interactome.
>
> •	A non-temporal, GRU-style layer-wise fusion gate that stabilizes attention propagation on noisy biological graphs and differs fundamentally from both prior approaches in this domain and the predominantly temporal uses of GRUs in other domains.
>
> •	Two fusion designs: (1) Embedding-Level EmbedFusion, which introduces a mixed-random-walk–plus–disease-set-to-vec embedding, and (2) Channel-wise GatedFusion, which adaptively integrates independently processed GGAT channels. Neither design has been explored in prior work.
>
> We have updated the manuscript to more clearly describe the architectural design, and we have added ablation studies (Table 2 and 3) demonstrating that each component—GAT, gating, and fusion—contributes meaningfully to the overall performance.
>
> **Response to weakness #2:**
>
> Thank you for this suggestion. We agree that the basic forms of GAT and the prediction layer are well known.
>
> Although our gating module adopts the GRU functional form, it is not the original GRU used for temporal recurrence. In GGAT, the gate operates across network layers rather than across time. The new information entering the gate comes from the current GAT neighborhood aggregation, while the stored information represents layer-wise accumulated context for each node. Moreover, the reset gate modulates how much of the new attention-derived signal is allowed through, rather than filtering the previous hidden state as in a temporal GRU. The formulas are therefore necessary to clarify how this gating mechanism is repurposed to regulate attention flow in a multi-layer graph model rather than to model temporal sequences.
>
> Thank you for this helpful comment. We have streamlined the math in the main text by replacing the standard GAT and BCE equations with brief descriptions, while moving the explicit formulas to an appendix for completeness. We retain only the equations that are necessary to specify our design of GGAT and fusion strategies, which we believe are important for clarity and reproducibility.

---

> ### Author Response · Authors · 2025-11-21
> **Response to Reviewer Xd2g (2)**
>
> **Response to weakness #3:**
>
> We would like to clarify that all existing interactome-based comorbidity prediction methods, including TSPE and prior baselines, rely exclusively on graph-derived biological signals, namely protein–protein interactions and disease–protein associations. Our framework follows the same setting to ensure a fair and consistent comparison.
>
> Pretrained single-entity embeddings such as ESM and BioBERT represent different biological modalities: ESM is a protein language model that captures sequence-level biochemical and structural features, while BioBERT encodes semantic information from biomedical texts (e.g., PubMed abstracts, MeSH descriptions), introducing external knowledge that is not part of the benchmark protocol. Incorporating these modalities would therefore change the problem setting and introduce information not used by any interactome-based baselines, leading to an unfair comparison with TSPE.
> That said, we agree that integrating heterogeneous biological modalities (e.g., sequence, structure, or text) is an interesting direction. It is indeed worth investigating whether such fine-grained biochemical or lexical features could meaningfully benefit comorbidity prediction, which primarily depends on disease-module relationships among sets of proteins. However, incorporating these modalities would require substantial additional work and is beyond the scope of the present study, which aims to establish a strong and interpretable framework under the standard interactome-based comorbidity prediction protocol.
>
> We appreciate the suggestion and have included this potential direction in the future-work discussion in the conclusion section.
>
> **Response to weakness #4:**
>
> The HI interactome was chosen for consistency with TSPE and prior work, allowing direct comparison with the current state of the art and other baselines. We agree that broader evaluation would be valuable. However, at present HI remains the only established benchmark for interactome-based disease comorbidity prediction, which limits the availability of alternative datasets for systematic evaluation. In the revised manuscript, we now discuss this as a limitation, and in the future-work discussion we outline larger and more up-to-date human interactome datasets that we plan to explore in subsequent extensions of this work.
>
> **Response to weakness #5:**
>
> We appreciate the reviewer’s request for greater clarity. In the revised manuscript, we have added the missing experimental details to the appendix, including:
>
> •	description of the data-splitting protocol used for the 10-fold cross-validation,
>
> •	the training procedure and hyperparameter settings,
>
> •	the implementation environment used for running the experiments.
>
> We have also updated the Materials section (Section 4.1) to explicitly note that all implementation details are provided in the appendix.
>
> **Response to weakness #6:**
>
> Thank you for this feedback. In the revised manuscript, we have added a Limitations paragraph to the Conclusion section. This new paragraph outlines the constraints of the current study, including:
>
> • evaluation is restricted to the HI interactome and the RR dataset, which limits systematic assessment of generalizability across alternative interactomes;
>
> • PPI networks are incomplete, undirected, and not context-specific, meaning they cannot represent condition-dependent or directional biological relationships;
>
> • comorbidity labels are derived from relative-risk estimates, which are imperfect and influenced by demographic and clinical factors, although RR remains the established ground truth for interactome-based comorbidity prediction;
>
> • the fusion module applies only to node-aligned learned representations that map one-to-one onto protein nodes and is not a general multimodal integration framework.

---

### Official Review · Reviewer_szLs · 2025-10-31

**Soundness:** 3
**Presentation:** 3
**Contribution:** 2
**Rating:** 2
**Confidence:** 2

**Summary:**

The authors propose a multi–channel fusion-based Gated Graph Attention Network (GGAT) mechanism to predict disease comorbidity. The problem addressed is impactful and challenging. Authors use a number of graph attention networks and associated data representation. However, some of the complexity of the task are yet to be addressed. Survey of related work that appears in computational biology literature is incomplete.

**Strengths:**

1) Disease comorbidity prediction is a mature but challenging problem. Hence it is still relevant to the research community.
2) Paper introduces a Gated transformer based architecture, which regulates the information flow with respect to their importance.
3) Authors integrate protein-protein interaction network topology and disease-protein association connectivity for better feature learning in comorbidity prediction.

**Weaknesses:**

1) Disease comorbidity is a challenging problem, as comorbidity can never be explained through one or two types of biological or clinical modality. PPI interaction itself is a quite sparse, context dependent, and undirected network that fails to capture the complete etiological landscape.
2) Comorbidity arises from multilayer determinant features like genetic, epigenetic, transcriptomic, metabolic, immune, environmental stress and clinical. Apart from that longitudinal EHR data combined with pathway, regulatory, and phenotypic networks are crucial to incorporate with cohort specific PPI data.
3) The rationale of the proposed work is so naive as the incompleteness in PPI and over-represented well studied disease-associated proteins only give bias to the outcomes.
4) Relying on a Gated mechanism to “let-in” the most crucial features from the past information risks encoding temporal and modality bias: as the learned attention may overweight that are plentiful in the training graph, i.e., typically PPI topology and disease-associations while ignoring the heterogeneous context specific determinants of comorbidity (e.g., age, sex, medication, immune-state etc.).
5) As comorbidity is multi-causal and context dependent, so attention restricted to PPI and its well known disease association can yield spuriously high edge weights for well-studied protein-disease associations. To mitigate these, the authors should introduce multi-channel fusion on heterogeneous modalities explicitly, limit the attention to reflect recency context and causal plausibility.
6) The learned gate over multiple modality can well decide the crucial information to let in the model. That lacks the proposed work.
7) Node2vec embedding with protein-disease association mostly encodes PPI topology and disease-wise co-occurrence, thereby overemphasizing the popular hubs and homophily while ignoring context, directionality, clinical, and environmental stressors - an approach is too naive for the mechanistic complexity of disease comorbidity. Hence, making such embeddings too naive to predict comorbidity beyond superficial network co-localization.
8) The proposed work appears to equate comorbidity “robustness” with relative-risk weighted links between disease pairs. However, RR is a marginal, prevalence dependent association that (i) is even confounded by multi-dimensional factors like age, sex, treatments etc. (ii) is also vulnerable to Berkson’s bias, Simpson’s paradox, (iii) it lacks directionality and temporality (onset order lag), and (iv) unstable across hospitals, period, and sub-populations. Using RR as the sole input network risks mistaking exposure patterns for biology and overstates pair “robustness” without causal adjustment or tune-to-event modeling.

**Questions:**

1) Disease comorbidity prediction is a well studied problem, which is not only associated with genetic mutations and only PPI information never able to give insight to address the fact too. Even PPI information and disease-protein pairs never give insights about the proper disease progression mechanisms. Authors should provide a discussion on these challenges.

2) From line 75-77 what is the other information integrated and how?

3) Under sec. 2.3, there are plenty of approaches done on comorbidity prediction and analysis like tensor factorisation, gene co-expression based analysis. But the authors are quite naive into their literature survey. The literature survey should be expanded.

4) Under sec. 3.1 how the authors design the attention pooling on which information perspectives?

5) How the "simple node2vec random path" and "protein-disease" association set can give most effective information in embedding learning, as it only provides PPI based topological and disease stratified protein information, which quite naive to define the complex trait of comorbid diseases?

6) from line 233-244, how do the authors think of letting entry be the most important feature to incorporate from the past information using the GATED approach ? As, there is plausible bias present in their embeddings to get learned from past (only) protein topological sequences and disease associations, as comorbidity depends on multiple heterogeneous factors.

7) In line 285. what are the different biological signals?

8) node2vec treats graph as a homogeneous one, hence it captures the PPI-based node-node connectivity information only, which is never a confirmatory way to give insights about novel comorbid diseases.

9) In line 319. how do the authors explain the reliability and robustness of the complementary information?

10) How do authors define the disease-disease pair robustness with relative risk scores only?

11) Heterogeneous biological signals can never be accumulated from PPI and disease association information only. Hence, the comorbidity prediction is supposed to be highly biased to Protein connectivity information only.

**Details Of Ethics Concerns:**

No ethics issue

---

> ### Author Response · Authors · 2025-11-21
> **Response to Reviewer szLs (1)**
>
> We sincerely thank the reviewer for the time and the detailed comments and questions. We address each point in detail below.
>
> **Response to weakness #1:**
>
> We agree that comorbidity is a multi-factorial phenomenon and that PPI-based interactomes capture only a subset of the underlying biological complexity. This limitation is shared by all interactome-based comorbidity prediction frameworks.
> However, our study focuses on network-based comorbidity prediction using the human interactome (PPI), which is the standard setting adopted in all prior interactome-based comorbidity models. Our goal is not to provide a full etiological or causal explanation of comorbidity, but to evaluate whether gated graph-attention–based message passing on the interactome can more effectively extract useful relational information compared to existing baselines.
>
> Our results show that, despite the limitations of the PPI mentioned above, the interactome still contains meaningful structural signals that can be effectively leveraged by our GGAT architecture and multi-channel fusion design.
>
> Thank you for the comment. We have clarified this scope and described the limitations of the PPI in the revised “Limitations” section.
>
> **Response to weakness #2:**
>
> We agree that comorbidity is influenced by many biological and clinical determinants. These modalities can indeed provide valuable complementary information.
>
> However, the focus of this study remains interactome-based comorbidity prediction, where we aim to evaluate whether improved graph message passing can better extract the information present in the network, rather than to construct a full multi-omics or EHR-integrated framework.
>
> Although our multichannel fusion architecture is flexible and already capable of incorporating additional learned node representations, extensions that combine heterogeneous network structures (e.g., pathway, regulatory, phenotypic, or EHR-derived graphs), while scientifically valuable, would require substantial methodological development and therefore fall outside the scope of this study. Future work may further extend this architecture to incorporate such heterogeneous biological or clinical networks once appropriate integration strategies are available. We have clarified this scope and noted these potential extensions in the revised manuscript.
>
> **Response to weakness #3:**
>
> We agree that PPI networks are incomplete and that proteins associated with well-studied diseases tend to be more extensively mapped, which can introduce bias. These limitations are well recognized in the domain and apply equally to all interactome-based comorbidity prediction frameworks.
>
> Importantly, Menche et al. (Science, 2015) demonstrated that even an interactome that is both incomplete and affected by mapping bias still retains sufficient structural information to identify disease modules and predict disease–disease relationships, including comorbidity. Our study follows this established setting.
>
> Our objective in this work is not to claim completeness of the PPI nor to provide an unbiased etiological model of comorbidity, but rather to evaluate whether our GGAT architecture can more effectively extract relational signals from the available interactome compared with existing baselines.
>
> Our results further support this view: despite the limitations of the current interactome, GGAT models achieve consistently strong performance across all benchmark metrics, indicating that meaningful structural information is indeed present and can be effectively captured for disease comorbidity prediction.
>
> Moreover, as interactome resources continue to expand and become more complete, the specific outcomes of comorbidity prediction will naturally change. However, methods that can more effectively capture the topological structure of the PPI, such as GGAT, are likely to remain advantageous when retrained on updated networks because they are designed to make use of relational information rather than depending on artifacts of a particular dataset.
>
> We have clarified these limitations in the revised “Limitations” section.
>
> [1] Menche, Jörg, Amitabh Sharma, Maksim Kitsak, Susan Dina Ghiassian, Marc Vidal, Joseph Loscalzo, and Albert-László Barabási. "Uncovering disease-disease relationships through the incomplete interactome." Science 347, no. 6224 (2015): 1257601.

---

> ### Author Response · Authors · 2025-11-21
> **Response to Reviewer szLs (2)**
>
> **Response to weakness #4:**
>
> We agree that comorbidity is influenced by heterogeneous clinical and demographic factors. However, none of these are represented in the interactome. They are therefore outside the scope of all interactome-based comorbidity prediction frameworks, not only ours. As with all interactome-based comorbidity frameworks, our model operates strictly on PPI structure and disease–protein associations, and therefore cannot encode temporal or clinical biases that are not part of the input.
>
> The gating mechanism in GGAT does not introduce temporal or clinical modality bias. It is a layer-wise feature update and is not a temporal or cross-modality selection mechanism. Moreover, no temporal, demographic, or patient-level modalities are available to the model in this interactome-based setting, so the gate cannot overweight information that the model never receives.
>
> Our ablation studies (Tables 2 and 3) show that gated GGAT consistently outperform non-gated variant GAT, indicating that the gating mechanism enhances the extraction of meaningful network structure rather than encoding unintended biases.
>
> We have clarified this scope and its implications in the revised manuscript.
>
> **Response to weakness #5:**
>
> We agree that comorbidity is multi-causal and depends on clinical, demographic, immunological, and environmental context. However, these context-specific variables are not represented in the interactome and therefore lie outside the scope of all interactome-based comorbidity prediction frameworks, including ours. Our model, like prior work, operates strictly on protein–protein interaction structure and disease–protein associations.
>
> The concern that attention mechanisms may assign spuriously high weights to well-studied proteins typically arises when a model has access to heterogeneous modalities or temporal information. In our setting, however, the attention mechanism operates only on two types of input embeddings: structural embeddings derived from the PPI network and embeddings derived from disease membership sets. The model does not receive any clinical, demographic, temporal, or other heterogeneous modalities. Therefore, it cannot favor recent events, temporal context, or cross-modality signals. The attention mechanism can only reflect the structural patterns in the interactome and the information already encoded in the input embeddings.
>
> Regarding heterogeneous modality fusion, our framework includes a multichannel fusion module only at the node-aligned feature level, meaning it applies exclusively to learned representations that map one-to-one onto protein nodes. It is not a general multimodal integration framework. Extending the architecture to incorporate additional biological or clinical modalities would require heterogeneous datasets that are not available in the current interactome-only benchmark and would fall outside the scope of this study. The current experimental setting is therefore consistent with all existing interactome-based comorbidity prediction studies.
>
> **Response to weakness #6:**
>
> We appreciate the suggestion regarding gating over multiple modalities. However, heterogeneous biological or clinical modalities are not available in the current interactome-based benchmark, and this limitation applies to all prior interactom-based comorbidity prediction frameworks, not only ours. Our model therefore operates within the standard single-modality setting defined by the human interactome and disease–protein associations.
>
> Because the dataset provides no heterogeneous modalities such as clinical, demographic, temporal, or multi-omics signals, multimodal gating is not applicable in this setting. Introducing such a mechanism would therefore fall outside the scope of this study.
>
> We agree that incorporating additional biological or clinical modalities and exploring multimodal gating is a valuable direction. This requires heterogeneous data sources beyond the current interactome benchmark. As noted in the revised manuscript, this is mentioned only as a potential extension for future work.

---

> ### Author Response · Authors · 2025-11-21
> **Response to Reviewer szLs (3)**
>
> **Response to weakness #7:**
>
> node2vec reflects only the intrinsic connectivity through random-walk neighborhoods; it does not artificially overweight highly connected proteins beyond what already encodes. The disease-association embedding captures co-membership within disease gene sets, but it does not reflect frequency-based or clinical co-occurrence. Each protein appears only once per disease, so the embedding encodes disease-module structure rather than amplifying hubs or popularity bias.
>
> It is important to clarify that additional determinants of comorbidity, such as biological directionality, cellular context, are not part of the current interactome-based benchmark. No method evaluated in this setting has access to such modalities. These factors are therefore not “ignored”; they are simply absent from the dataset on which all existing work in this domain is based.
>
> Crucially, Menche et al. (Science, 2015) demonstrated that even an incomplete and context-agnostic interactome contains sufficient structural information to identify disease modules and analyze disease–disease relationships, including comorbidity. Our study follows this established setting.
>
> Within our framework, node2vec or disease-association embeddings serve as channel input, GGAT then learns task-relevant higher-order representations through gated graph-attention layers trained on known comorbidity, which prevents the learned representations from merely overemphasizing hubs or homophily. The attention and gating mechanisms adaptively prioritize useful relational signals rather than inheriting biases from the raw embeddings.
>
> Furthermore, the comorbidity labels used for training originate from clinical data and relative-risk estimates (Menche et al., 2015), ensuring that our prediction task reflects clinically meaningful associations rather than artifacts.
>
> The strong performance of GGAT using only PPI-derived structure and disease–protein associations indicates that the n2v and disease association embeddings still retain enough relevant info for a good learner to learn. Despite the limitation of the embeddings, the fact that our work outperforms the state-of-the-art is a testament that previous methods didn’t fully exploit what these embeddings and PPI networks have to offer for comorbidity prediction. While our study does not claim to achieve the upper bound of what can be learned from the interactome, but at least we succeeded in pushing the existing boundary.
>
> Although the representations learned by GGAT from the PPI network contain useful task-relevant information for comorbidity prediction, they are not directly interpretable in terms of cellular context or clinical factors, as such modalities are absent from the dataset. We expect that incorporating additional biological or clinical modalities in future work will further strengthen comorbidity prediction and enhance the biological relevance of the learned representations.
>
> We have clarified the scope, limitations, future work in the revised manuscript.
>
> [1] Menche, Jörg, Amitabh Sharma, Maksim Kitsak, Susan Dina Ghiassian, Marc Vidal, Joseph Loscalzo, and Albert-László Barabási. "Uncovering disease-disease relationships through the incomplete interactome." Science 347, no. 6224 (2015): 1257601.
>
> **Response to weakness #8:**
>
> We agree that relative-risk (RR)–based disease associations have well-known limitations; However, these limitations are inherent constraints of the RR labels provided in the benchmark dataset, and they apply to all interactome-based comorbidity prediction frameworks.
>
> Importantly, our work does not equate comorbidity “robustness’’ with RR itself, nor do we use RR values as model inputs. RR serves only as ground-truth labels for supervised learning in this domain, following the standard setting established in prior interactome-based studies. Our model operates solely on the PPI network and disease–protein associations and does not attempt causal inference or temporal modeling over clinical variables, which are not available in the benchmark.
>
> Similar situations occur widely in computational biology. For example, structure-prediction methods (from molecular dynamics to AlphaFold) rely on experimentally determined X-ray structures as ground truth, even though these structures also contain measurement noise and resolution limits. RR-based comorbidity estimates play an equivalent role in interactome-based comorbidity modeling.
>
> We have added this clarification and limitations of RR in the revised Limitations section.

---

> ### Author Response · Authors · 2025-11-21
> **Response to Reviewer szLs (4)**
>
> **Response to Q1:**
>
> We agree that PPI and disease–protein associations alone cannot capture the full biological or clinical determinants of comorbidity, nor do they provide mechanistic explanations of disease progression. These challenges are intrinsic to the benchmark used in all interactome-based comorbidity studies. Our work does not attempt to provide mechanistic or causal insight but evaluates whether improved message passing via gated graph attention can more effectively extract the relational structure that is present in the interactome. We have discussed this limitation in the revised manuscript.
>
> **Response to Q2:**
>
> Thank you for the question. The “biological information sources” referred to in lines 75–77 denote the two feature sources used in our model: connectivity-based protein embeddings and disease–protein association embeddings, as well as any additional node-aligned features that could be incorporated in future extensions.
>
> We have revised the manuscript to clarify that framework currently integrates only two available information sources.
>
> **Response to Q3:**
>
> To our knowledge, most tensor-factorization and gene–co-expression studies focus on general disease analysis (e.g., phenotyping or module detection) rather than on disease–disease comorbidity prediction. Among existing tensor-factorization approaches, the work of Biswas et al. is the only one we are aware of that directly targets comorbidity prediction, and it does not rely on the interactome. We are also not aware of gene co-expression–based models that explicitly formulate comorbidity prediction.
>
> We appreciate the reviewer’s comment. We have revised Section 2.3 accordingly to separate interactome-based comorbidity models, non-interactome comorbidity approaches, and single-disease analysis methods, thereby clarifying the position of interactome-based comorbidity modeling within the broader landscape of computational disease analysis.
>
> [1] Biswas, S., Mitra, P. and Rao, K.S., 2019. Relation prediction of co-morbid diseases using knowledge graph completion. IEEE/ACM Transactions on Computational Biology and Bioinformatics, 18(2), pp.708-717.
>
> **Response to Q4:**
>
> We thank the reviewer for the question. The phrase attention pooling was the earlier name of the module that is now referred to as adaptive pooling. The naming was unified to avoid confusion with true attention-based pooling mechanisms. The adaptive pooling mechanism used in GGAT is described in Section 3.1. We have corrected this wording in the revised manuscript and ensured consistent terminology throughout.
>
> **Response to Q5:**
>
> We thank the reviewer for the comment. This point is already addressed in our response to Weakness #7, where we clarify why the embeddings used in the HI benchmark remain appropriate for this setting and explain why the concern does not apply under the benchmark’s constraints. Please refer to that discussion.
>
> **Response to Q6:**
>
> We thank the reviewer for the question. We clarify that our method does not use temporal information, and the term “past” does not refer to biological or clinical history. In lines 233–244, the phrases “past states” and “previous hidden state” follow the standard GRU terminology and refer only to the layer-wise hidden state h_i^((l) )from the previous GNN layer. The gate therefore regulates information flow across layers, not across time.
>
> The gated mechanism cannot introduce additional heterogeneous factors beyond the embeddings themselves, since the HI benchmark provides only PPI topology and disease–protein membership. The gate simply stabilizes GAT aggregation on noisy interactome graphs and prevents over-smoothing; it does not amplify biological bias or prioritize any temporal “past” information. Please refer to the detailed discussion in Weakness #4.
>
> **Response to Q7:**
>
> Thank you for the question. The phrase “different biological signals” refers specifically to the two sources of interactome-derived information used in this study, protein connectivity and disease–protein association information, as stated at the end of the sentence. To avoid ambiguity, we have revised the corresponding sentence in the manuscript to explicitly mention these two signals.
>
> **Response to Q8:**
>
> We thank the reviewer for the comment. This concern is already addressed in our response to Weakness #7. In that response, we clarify that the HI benchmark introduced by Menche et al. (2015) demonstrated that PPI-derived information contains sufficient structural signals for disease–disease analysis, and all prior interactome-based comorbidity studies have followed this setting. Our aim is not to argue that such embeddings capture the biological mechanisms of comorbidity. Instead, we evaluate whether our model can make more effective use of the available interactome information than existing approaches. For more details, please see our full response to Weakness #7.

---

> ### Author Response · Authors · 2025-11-21
> **Response to Reviewer szLs (5)**
>
> **Response to Q9:**
>
> We thank the reviewer for the question. The complementary information refers to the two independently derived learned representations from the two channels, GGAT-Connect and GGAT-Disease. Each channel is trained with a different type of input embedding: Node2Vec for connectivity-based information and Label2Vec for disease–protein association information.
>
> As schematically illustrated in Figure 1, the node sequences generated by random walks (Node2Vec) and those derived from disease–protein association sets follow different statistical patterns. In addition, random-walk sequences encode graph connectivity, whereas disease–association sequences encode disease-module membership. Taken together, these differences cause the resulting GGAT-Connect and GGAT-Disease representations to capture complementary rather than redundant information.
>
> Their reliability and robustness are supported empirically. Each single-channel model (GGAT-Connect and GGAT-Disease) achieves strong predictive performance on its own, indicating that each information source is meaningful. Furthermore, integrating the two channels through the GatedFusion model consistently yields additional performance gains (Table 1), demonstrating that the complementary information contributes useful signal for comorbidity prediction.
>
> **Response to Q10:**
>
> We thank the reviewer for the question. This point is already addressed in our response to Weakness #8, where we clarify that we do not define disease–disease pair “robustness’’ using relative risk. RR values are used only as benchmark labels, consistent with prior interactome-based studies, and our model does not rely on RR as a mechanistic or causal measure. Please refer to our detailed response to Weakness #8.
>
> **Response to Q11:**
>
> We thank the reviewer for the comment. This concern is already addressed in our response to Weakness #7, where we clarify that although the benchmark provides only PPI connectivity and disease–protein associations, GGAT does not become biased toward protein connectivity alone. The gating and attention mechanisms learn task-relevant higher-order relational patterns rather than amplifying raw topological bias. Please refer to the detailed discussion in Weakness #7.

---

> > ### Comment · Reviewer_szLs · 2025-11-26
> >
> > We accept your argument that the method may be extended to other heterogeneous networks. However, effectiveness of such a technique can only be established after the experiments are performed. Often the effectiveness of a technique extends to larger and heterogeneous networks. The work is promising but to a biology audience more concrete results and insights are sought.

---

> > > ### Author Response · Authors · 2025-11-26
> > >
> > > We thank the reviewer for the encouraging feedback and for recognizing the promise of our approach. We appreciate the constructive suggestion about evaluating the method on larger or heterogeneous biological networks.
> > >
> > > At this stage, we have established a complete and reproducible interactome-based comorbidity prediction pipeline that follows the standard benchmark setting used in prior work. Comparing our model within this established benchmark is the standard and accepted way to evaluate whether a new pipeline provides improved effectiveness over existing approaches in this domain.
> > >
> > > As noted in the manuscript, extending the framework to incorporate additional heterogeneous biological networks or experimental validation would require data beyond what is available in the current benchmark, and therefore lies outside the scope of this study.
> > >
> > > That said, we agree that such extensions represent valuable next steps. We will take the reviewer’s constructive feedback into account as we continue to develop this line of work, including exploring integration of richer biological networks and pursuing collaborations that may enable future biological validation.
> > >
> > > We sincerely appreciate the reviewer’s insight and positive assessment.

---

### Official Review · Reviewer_wmgg · 2025-10-31

**Soundness:** 2
**Presentation:** 3
**Contribution:** 2
**Rating:** 2
**Confidence:** 4

**Summary:**

This paper proposes a Gated Graph Attention Network (GGAT) framework for disease comorbidity prediction, aiming to address limitations of the state-of-the-art (SOTA) Graph Transformer with Subgraph Positional Encoding (TSPE) method. The framework’s core design includes three key innovations compared to TSPE: applying attention over local neighbors (instead of global pairwise attention) for more biologically meaningful feature aggregation, integrating a gating mechanism to adaptively regulate information flow and enhance representation learning, and introducing a multichannel fusion strategy to combine connectivity-based and disease association-based embeddings. Experiments show the effectiveness of the proposed model.

**Strengths:**

1. The paper has a clear structure and is easy to read

2. This paper proposes a Gated Graph Attention Network (GGAT) framework for disease comorbidity prediction. The overall model architecture seems reasonable.

**Weaknesses:**

1. Core components of GGAT lack originality. Local neighbor attention is a defining feature of standard GATs, the GRU-based gating mechanism replicates prior work on Gated GCN, and multichannel fusion adapts existing fusion techniques from GNNs and computer vision. The manuscript does not introduce innovations to these components (e.g., disease-specific attention weighting, task-aware gating) or provide theoretical justification for their combination beyond incremental performance gains.

2. The experiments are not convincing enough. The authors only compare to TSPE variants, omitting critical baselines like standard GNNs (GCN, GraphSAGE), recent comorbidity-focused models, and vanilla GAT. The baselines in 2025 should be compared.

3. A key motivation for GGAT is addressing TSPE’s high computational cost from positional encodings. Can you provide quantitative comparisons of computational overhead (FLOPs, training/inference time, memory usage) between GGAT (especially GatedFusion with dual channels) and TSPE on the RR1 dataset? Does GGAT’s efficiency scale to larger interactomes or datasets?

**Questions:**

Please see the weaknesses

---

> ### Author Response · Authors · 2025-11-20
> **Response to Reviewer wmgg (1)**
>
> We sincerely thank the reviewer for the thoughtful feedback. Your comments have helped us refine the presentation and expand the experiments. We have carefully revised the manuscript and address each point below.
>
> **Response to weakness #1:**
>
> We would like to clarify that our work is application-driven rather than focused on introducing new GNN primitives. Modeling disease-specific protein sets and disease–disease relationships—tasks fundamentally different from standard node-level GNNs. To our knowledge, no prior method jointly uses local attention, gated propagation, and multichannel or mixed random-walk–plus–disease-set embeddings to learn disease-conditioned protein representations and derive set-level disease embeddings.
>
> Recent SOTA transformer-based comorbidity models demonstrate the utility of attention, but their global attention mechanism is less suited for capturing local interactome structure. We therefore adopt GAT, which provides localized and edge-aware attention tailored to PPI neighborhoods. However, because the interactome is noisy, attention alone cannot ensure stable multi-layer propagation. To address this, we incorporate a layer-wise GRU gating mechanism to stabilize information flow across GAT layers. Our design thus begins from GAT—motivated by the domain’s SOTA method—and extends it with task-specific gating to meet the requirements of comorbidity prediction.
>
> And our gating mechanism is an architectural innovation in this setting: it is not the temporal GRU commonly used in GNN or computer-vision models, but a GRU-style layer-wise gate specifically adapted to stabilize attention propagation on sparse and noisy biological graphs.
>
> Specifically, the originality lies in the combination and problem-specific adaptation:
>
> •	A fundamentally different architectural design from prior interactome-based comorbidity prediction methods, explicitly modeling disease-specific protein sets and disease–disease interactions rather than relying on node-level GNN embeddings, and advancing the state of the art in prediction.
>
> •	Local biological attention via GAT. To address the limitation of SOTA method TSPE, which relies solely on global pairwise attention, we leverage graph attention over local PPI neighborhoods to capture biologically meaningful and topologically coherent interactions within the interactome.
>
> •	A non-temporal, GRU-style layer-wise fusion gate that stabilizes attention propagation on noisy biological graphs and differs fundamentally from both prior approaches in this domain and the predominantly temporal uses of GRUs in other domains.
>
> •	Two fusion designs: (1) Embedding-Level EmbedFusion, which introduces a mixed-random-walk–plus–disease-set-to-vec embedding, and (2) Channel-wise GatedFusion, which adaptively integrates independently processed GGAT channels. Neither design has been explored in prior work.
>
> We have updated the manuscript to more clearly describe the architectural design, and we have added ablation studies (Table 2 and 3) demonstrating that each component—GAT, gating, and fusion—contributes meaningfully to the overall performance.
>
> **Response to weakness #2:**
>
> We appreciate the reviewer’s suggestion regarding the need for broader baseline comparisons. In the revised submission, we have expanded our evaluation to include four levels of comparison: (1) Recent comorbidity-focused baselines: GE, BSE, and TSPE, which together represent all currently published interactome-based comorbidity prediction methods (Table 1).
> (2) Standard GNN baselines: GCN, GraphSAGE, and vanilla GAT (Tables 2 and 3).
> (3) Gated vs. non-gated ablations: Paired gated and non-gated variants of the same GNN backbones under identical input settings (Tables 2 and 3).
> (4) Channel-level ablations: Connectivity-only, disease-only, and fusion variants (Table 1), demonstrating the necessity of integrating heterogeneous biological information.
>
> These additions address the reviewer’s concern and show that GGAT achieves the strongest overall performance across all contemporary baselines (as of 2025).

---

> ### Author Response · Authors · 2025-11-20
> **Response to Reviewer wmgg (2)**
>
> **Response to weakness #3:**
>
> We are sorry for the confusion. Computational efficiency is not the primary motivation for introducing the Gated GAT with Multichannel Fusion framework. The key advantages over TSPE arise from three aspects. (1) GGAT uses local, edge-based attention rather than TSPE’s global Transformer attention, which computes pairwise interactions among all proteins in a disease or between a disease pair regardless of whether a true PPI edge exists. (2) GGAT avoids TSPE’s limitation in disease-information capacity: instead of injecting disease labels through a low-dimensional concatenation that restricts how much disease signal can be represented, GGAT uses a disease channel that can encode richer disease information. (3) GGAT provides flexibility that TSPE lacks. TSPE adds and concatenates all information into a single embedding, making it difficult to incorporate new biological features. In contrast, GGAT’s multichannel design allows additional signals to be added as new channels and fused through a lightweight gated module, without retraining existing channels or enlarging embedding vectors.
>
> For completeness, we clarify the computational implication referenced in the manuscript when describing the benefit of GGAT fusion for incorporating disease information. In TSPE, the subgraph positional encoding (SPE) is constructed by adding Laplacian positional encoding (LPE) to the connectivity embedding (M), and then concatenating a projected Graph Encoder Embedding (GPE). The GPE concatenation increases the Transformer input dimension. In the original TSPE implementation, only an 8-dimensional GPE projection was used; larger dimensions were avoided because increasing dGPE directly inflates the cost of each self-attention and encoder–decoder cross-attention block. In contrast, GGAT fusion does not combine all information into a single embedding. Instead, it processes connectivity and disease signals in two channels (GGAT-Connect and GGAT-Disease) with moderate dimensionality, and each GGAT layer performs sparse attention only over true PPI edges. This avoids TSPE’s dense attention over all node pairs within and between disease modules.
>
> Thank you for this helpful comment. We have updated the manuscript to clearly describe these motivations and distinctions.

---

### Note · Authors · 2026-01-20

I have read and agree with the venue's withdrawal policy on behalf of myself and my co-authors.